# Ontogeny independent expression of LPCAT2 in granuloma macrophages during experimental visceral leishmaniasis
Shoumit Dey [1,4], Jian-Hua Cao[2,4], Benjamin Balluff [2], Gaia Mazza [1], Helen Ashwin [1], Lesley Gilbert[3], Sally James[3], Adam A. Dowle [3], Grant Calder[3], Nidhi Sharma Dey [1], Peter O'Toole [3], Ron M. A. Heeren [2] ✉ & Paul M. Kaye [1] ✉

Granulomas are organized inflammatory lesions formed in response to persistent stimuli such as infections. Murine infection with *Leishmania donovani* results in granulomas in the liver, seeded by infected Kupffer cells, and serves as a well-defined model of infection-induced granuloma formation. The resolution of granulomatous inflammation requires dynamic shifts in immune-cell activation states, imposing metabolic demands. As mediators of cell signaling, lipid metabolism plays a key role in regulating inflammation and infection. How lipid changes are spatially linked to altered immune cell transcription remains unresolved. We performed a multimodal imaging analysis combining MALDI mass spectrometry, spatial and single cell transcriptomics, proteomics of flow-sorted macrophages and histopathology of *L. donovani* induced hepatic granulomas. Using this spatially-integrated approach, we identified LPCAT2-mediated membrane re-modelling of myeloid cells as a novel feature of these granulomas. Our study provides new insights into local immunometabolic changes associated with granuloma formation and macrophage activation.

Granuloma formation is a hallmark of many infectious and inflammatory diseases[1]. Well-organized granulomas characterize chronic infections like tuberculosis, where they contain bacteria but also provide a niche for persistence[2,3]. In sarcoidosis, aberrant granuloma responses cause organ damage[4]. Leprosy presents across a spectrum of granulomatous inflammation, with well-formed granulomas correlating with the self-limited tuberculoid form versus poorly organized inflammation in the disseminated lepromatous form[5]. In human and canine visceral leishmaniasis, granulomas are typically absent or poorly formed in clinical cases but demonstrable in asymptomatic individuals[6], suggesting a host protective role[7].

In mice, infection with *Leishmania donovani* results in the development of organized hepatic granulomas around infected Kupffer cells. The granulomatous response helps control parasite replication and resolve infection[8]. Surrounding a core of parasitized Kupffer cells other cell types accumulate, including CD4[+] T cells, CD8[+] T cells, monocytes, monocyte-derived macrophages (MDMs), and scant neutrophils under the influence of an evolving landscape of cytokine and chemokine expression. Within granulomas, the complex local cytokine environment in part dictates parasite survival or elimination[9], the latter leading over time to granuloma resolution (involution).

The formation and resolution of granulomatous inflammation requires dynamic shifts in immune cell populations and functional states, imposing significant metabolic demands. Upon activation, immune cells undergo metabolic reprogramming to support increased energy utilization and biosynthesis[10]. For example, inflammatory macrophages show increased glycolysis and decreases in TCA cycle and oxidative phosphorylation, and increased lipid metabolism[11]. As mediators of energy homeostasis and cell signaling, lipids and lipid metabolism are intertwined with immune cell activation[12]. In sterile disease, increased cholesterol and fatty acid (FA) biosynthesis, intracellular ceramide, and very-low-density lipoprotein (VLDL) secretion feedback to regulate immune responses[13]. For example, decreased expression of PLA2G7, a lipoprotein-associated phospholipase A2 (Lp-PLA2), was correlated with lower inflammation in calorie-restricted humans and improved metabolism in ageing mice[14]. Lp-

[1]York Biomedical Research Institute, Hull York Medical School, University of York, York, UK. [2]Maastricht MultiModal Molecular Imaging (M4I) Institute, Division of Imaging Mass Spectrometry, Maastricht University, Maastricht, the Netherlands. [3]Biosciences Technology Facility, Department of Biology, University of York, York, UK. [4]These authors contributed equally: Shoumit Dey, Jian-Hua Cao. ✉e-mail: r.heeren@maastrichtuniversity.nl; paul.kaye@york.ac.uk

PLA2 has also been shown to modulate macrophage M1 polarization in experimental autoimmune encephalitis[15]. These data suggest a micro-environment-specific, direct effect of lipid metabolism on macrophage activation[16,17]. Infectious diseases including TB and leishmaniasis have also been shown to result in altered lipid metabolism[18,19] though only rarely has this been analyzed in a spatial context[20,21].

In this study, we performed multimodal imaging that combines Matrix-assisted laser desorption/ionization mass spectrometry (MALDI) based lipid imaging (MSI), spatial transcriptomics (10x Visium), histo-pathology of hepatic granulomas (immunostaining), ex vivo proteomics, and flow cytometry to study the granuloma microenvironment during experimental *L. donovani* infection. Our integrated approach provides new insights into the immunometabolic changes associated with granulomatous inflammation in the hepatic microcosm and evidences the Lands Cycle[22] enzyme lysophosphatidylcholine acyltransferase 2 (LPCAT2) and its sub-strates as key components of granulomatous inflammation in the liver and in macrophage activation in vivo.

## Results

### Defining the cellular and molecular landscape of the L. donovani infected liver

Mature granulomas predominate in the liver of C57BL/6 mice infected for 28 days with *L. donovani*, minimizing heterogeneity due to granuloma evolution[23,24]. Using consecutive serial 7 μm sections processed for mor-phology (immunostaining), MSI and 10x Visium, we probed the tran-scriptomic and lipidomic landscape (Fig. 1a). To overcome the differences in spatial resolution between MSI (20 μm) and Visium (55 μm), we averaged 4–6 MSI pixels underlying each transcriptomic spot to generate a final integrated histopathological, transcriptomic and lipidomic map of liver niches at 55μm resolution, referred to as a spot (Fig. 1a). Using adjacent liver tissue from the same animals, we produced a combined single-cell atlas of liver cells from infected and naïve mice to allow deconvolution of cell type abundances within spatial spots. Granulomas had typical morphology (Fig. 1b) and were typically composed between 63-71 cells (Fig. 1c).

We first performed unbiased clustering of Visium RNA counts and lipid intensities separately (see "Methods") and visualized these spots in t-SNE space to find underlying structure within the two modalities (Fig. 1d). Upon projecting the lipid-guided clustering information onto the transcriptomics-led t-SNE space, Lipid_4 was found to be enriched within RNA_4 (center, Fig. 1d and Supplementary Fig. 1) and both were over-represented in infected mice (Fig. 1e). We then demonstrated that our unbiased RNA and lipid clusters showed spatial organization and RNA_4 and Lipid_4 overlaid granulomas defined histologically (Fig. 1f and Sup-plementary Fig. 2). In addition, RNA_7 also overlaid some granulomas (Fig. 1f). The top markers of RNA_4 included *Saa3* (Serum amyloid), *Ccl5* and *Lyz2* (Supplementary Fig. 3a). Lipid identities (described in Supple-mentary data 1) indicated that ether-linked PCs such as PC(O-36:3), PC(O-34:2), PC(O-38:5) were observed as the abundant lipid species in Lipid_4 (Supplementary Fig. 3b). We note that lipid clustering in some samples, particularly naïve tissues, displays geometric patterns in spatial plots (Supplementary Fig. 2, also see "Methods") due to incomplete spatial cov-erage. While both transcriptomic and lipidomic measurements indepen-dently demonstrate spatial organization, all subsequent downstream analysis was guided by RNA clusters.

### Delineating the single-cell transcriptomic landscape of granulomas

We performed scRNA-seq on adjacent liver tissue and manually annotated cell types (Fig. 2a, Supplementary data 2) to obtain cellular transcriptomes of infected liver non-parenchymal cells. From this, we determined the relative proportions of immune cells across samples (Fig. 2b) using marker genes from imputed clusters and additional canonical markers (Fig. 2c). In naïve mice, cell yields were $7–15 \times 10^6$ and B cells and T cells were represented at similar frequencies. In infected mice, cellularity increased ($6–10 \times 10^7$) with $Ifng^+CD4^+$ T cells becoming the dominant lymphocyte population (Fig. 2b

and Supplementary Fig. 4a–d). In infected liver, monocyte-derived mac-rophages (Lyz2Hi_MoMac, positive for *Ccr2*) were nearly twice as frequent than in naïve mice, whereas the proportion of Kupffer cells (ApoeHi_-Kupffer, positive for *Adgre1* / F4/80, *Clec4f*, and *Marco*) remained similar. Lyz2Hi_MoMac and ApoeHi_Kupffer cells were further distinguished by examining differentially expressed genes, which included some proin-flammatory genes such as *Tnf* and *Tlr2* (Fig. 2d and Supplementary Fig. 4e).

### Assessing lipid metabolism pathways in single cells through gene expression patterns

To determine if ApoeHi_Kupffer cells and Lyz2Hi_MoMacs had altered lipid metabolism, we compared the top macrophage upregulated genes from both populations with the Reactome (Metabolism of Lipids) gene list for *Mus musculus* (R-MMU-556833). 16 genes were common between Apoe-Hi_Kupffer cells and Reactome (Fig. 3a), including phospholipases or $PLA_2$ that convert phosphatidylcholines (PCs) to lysophosphatidylcholines (LPCs) such as *Pla2g4a* and *Pla2g15*. Additionally, genes downstream to arachidonic acid (AA) metabolism and involved in prostaglandin synthesis (*Ptgs1, Tbxas1*) via the metabolism of polyunsaturated fatty acids (PUFAs) were highly expressed[25]. *Lpcat2* which can incorporate acyl-CoAs to LPCs forming PCs (via the Lands Cycle) was also expressed in ApoeHi_Kupffer cells (Fig. 3b).

Similarly, the Lyz2Hi_MoMac and Reactome intersection (19 genes; Fig. 3c), also suggested enrichment of sphingolipid and AA metabolism pathways, along with the PC re-modeler *Lpcat2*. As PC hydrolysis can lead to AA release and LPCATs can incorporate acyl-CoAs such arachidonyl-CoA into PCs[26], we compared acyl-chain remodeling genes across cell types (Fig. 3d). Both ApoeHi_Kupffer cells and Lyz2Hi_MoMacs upregulated *Lpcat2*, *Pla2g4a*, *Pla2g15*, and *Pla2g7*. To confirm whether LPCAT2-mediated PC remodeling occurred on macrophages independent of their origin ($Ccr2^{hi}$ or $Adgre1^{hi}$) we calculated a score (using AddModuleScore in Seurat) for each cell using the genes *Lpcat2*, *Pla2g4a*, *Pla2g15* and *Pla2g7* (Fig. 3e). Highest scores were associated with the two macrophage popu-lations and with neutrophils (Fig. 3f). For these three myeloid cell popu-lations, high scores were associated with infected mice (Fig. 3g) and were observed in ~50% of each cell type (Fig. 3h). In contrast to more mature macrophages, $Ly6c^{hi}$ neutrophils / monocytes and $Itgax^+$ APCs had a very low LPCAT2 remodeling scores (Fig. 3f).

Further, there was a change in gene expression between high and low scoring cells, suggesting the former are more activated and/or inflammatory based on the expression of *C1qa-c*, MHC-II genes (*H2-Aa* and *H2-Eb1*), *Apoe*, *Cxcl9*, *Saa3* and *Cd5l* (Fig. 3i). Indeed, when LPCAT2/PL Remodeling score is correlated against an inflammatory (based on *Tnf*, *Nos2*, *Il6*, *Cd86* and *Tlr2*) or anti-inflammatory (based on *Il10*, *Tgfb1*, *Arg1*, *Il4*) we find remodeling score to correlate most highly with the inflammatory score (Supplementary Fig. 5a). Within myeloid cells with a high LPCAT2/PL remodeling score, *Lpcat2* positively correlated with genes related to pha-gocytosis, lipid processing, and immune / inflammatory responses, including *Pla2g7*, *Marcks, Sirpa, Mertk*, and *Fcgr4* (Fig. 3j).

To address whether *Lpcat2* was also upregulated in other tissues, we mined a publicly available single-cell dataset from *L. donovani* infected BALB/c mice[27]. *Csf1r*-expressing $Lpcat2^+$ cells were observed in the bone marrow and spleen of these infected mice (Supplementary Fig. 5b).

### Distinguishing granulomatous inflammation

We next used cell2location (see "Methods") to predict cell abundances within each RNA cluster. RNA_4 was enriched in macrophages, $CD4^+$ and $CD8^+$ T cells, NK cells, and neutrophils. In addition, RNA_0 and RNA_7 were also predicted to contain immune cells with T cell sub-types including Tregs and APCs respectively (Fig. 4a). Hepatocytes were most abundant in RNA_5 (hepatocytes1 and hepatocytes2) and in RNA_0 (hepatocytes2) (Fig. 4a). As immune cells were concentrated in RNA_0, RNA_4 and RNA_7 (Fig. 4a) but immune cells were also found in naïve mice (Fig. 2b), we combined these immune clusters and performed subclustering, identi-fying 5 sub-clusters (Sub0-4) which separated infected from naïve tissue

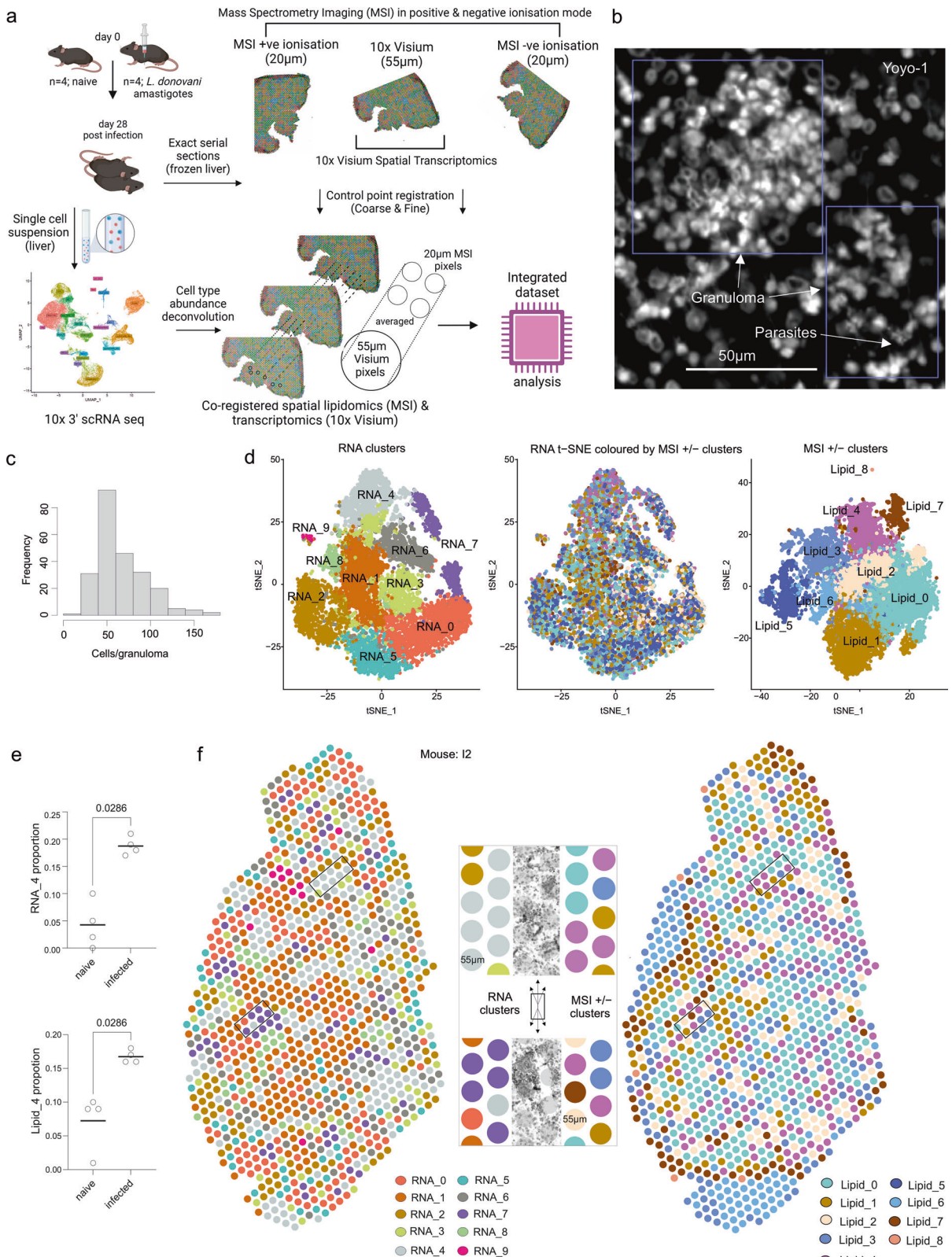

**Fig. 1 | Transcriptomic and lipidomic organization of immune granulomas.**
**a** Overall study plan (see "Methods" for details). **b** Granuloma morphology using nuclear stain (Yoyo-1, white). Parasites can also be observed in some granulomas. **c** Distribution of cell numbers per granuloma. **d** Clusters identified using top principal components separately for RNA and lipids are visualised in t-SNE space (left and right panels, respectively). Spots are visualised in transcriptomic t-SNE space colored by lipidomic cluster. **e** Proportional cluster composition of RNA_4 (top) and Lipid_4 (bottom) for naïve and infected mice. *P* values are reported using a Mann–Whitney test. **f** Spatial plot of representative infected tissue showing RNA clusters (left) and Lipid clusters (right). The center panel shows spots overlying tissue morphology (counterstained by DAPI).

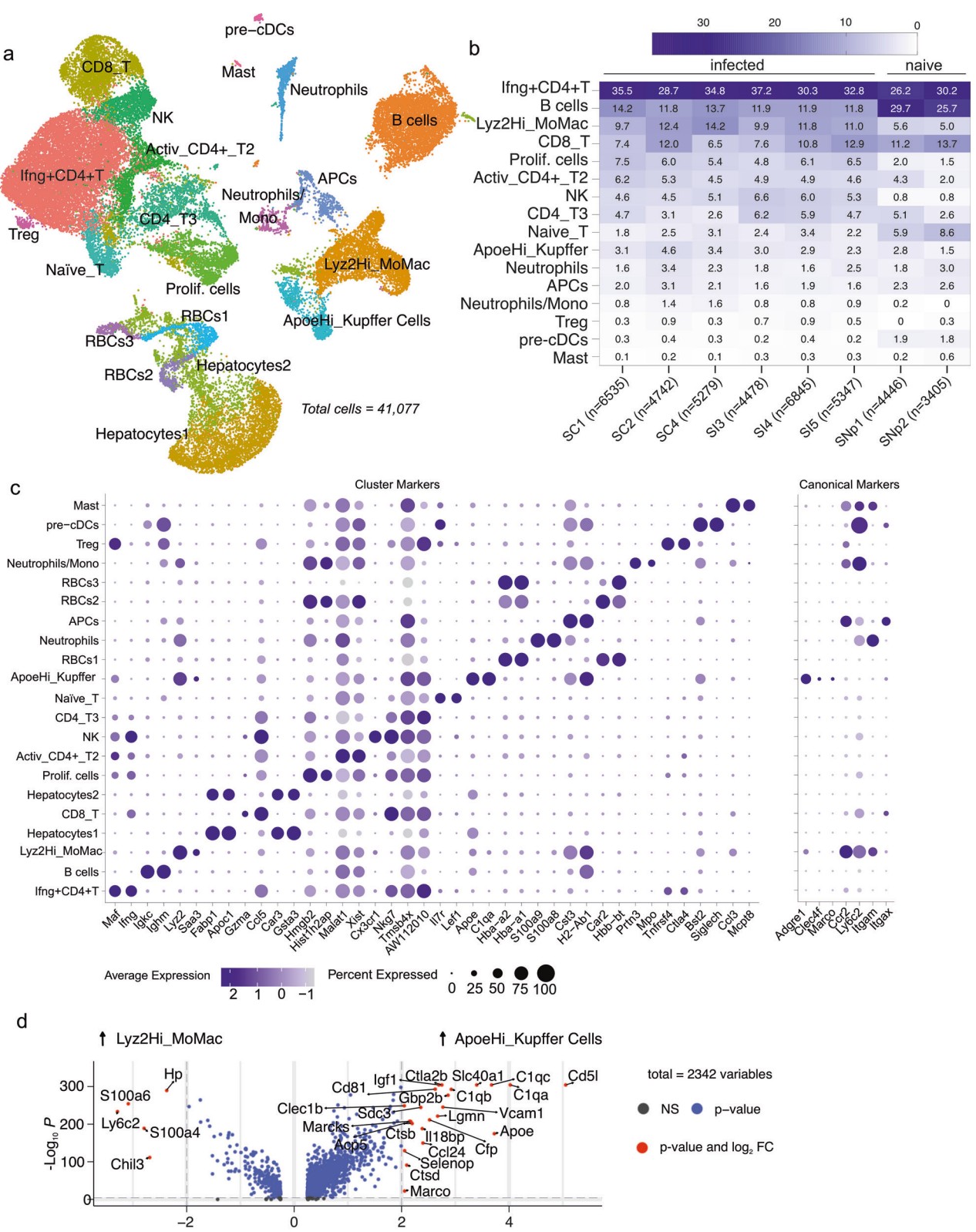

**Fig. 2 | Single-cell transcriptomic landscape of *L. donovani* infected liver.**
**a** Imputed cell types visualised in 2-dimensional UMAP space (axes not shown).
**b** Heatmap showing the frequencies of the single-cells across individual infected and naïve samples (*n* = 12; 6 infected (3 matched mice from spatial transcriptomics and 3 from an independent experiment) mice plus 2 pools of 3 naïve mice each). Samples names and number of cells sequenced are indicated on x-axis ticks. **c** Dot plot showing top 2 genes expressed by the imputed cell types. Dot size represents percentage of expressing cells. Color represents fold change. Canonical markers (right) discriminating cell types. **d** Volcano plot of differentially expressed genes between Kupffer cells and monocyte-derived macrophages. Bonferroni-corrected Wilcoxon-rank sum test.

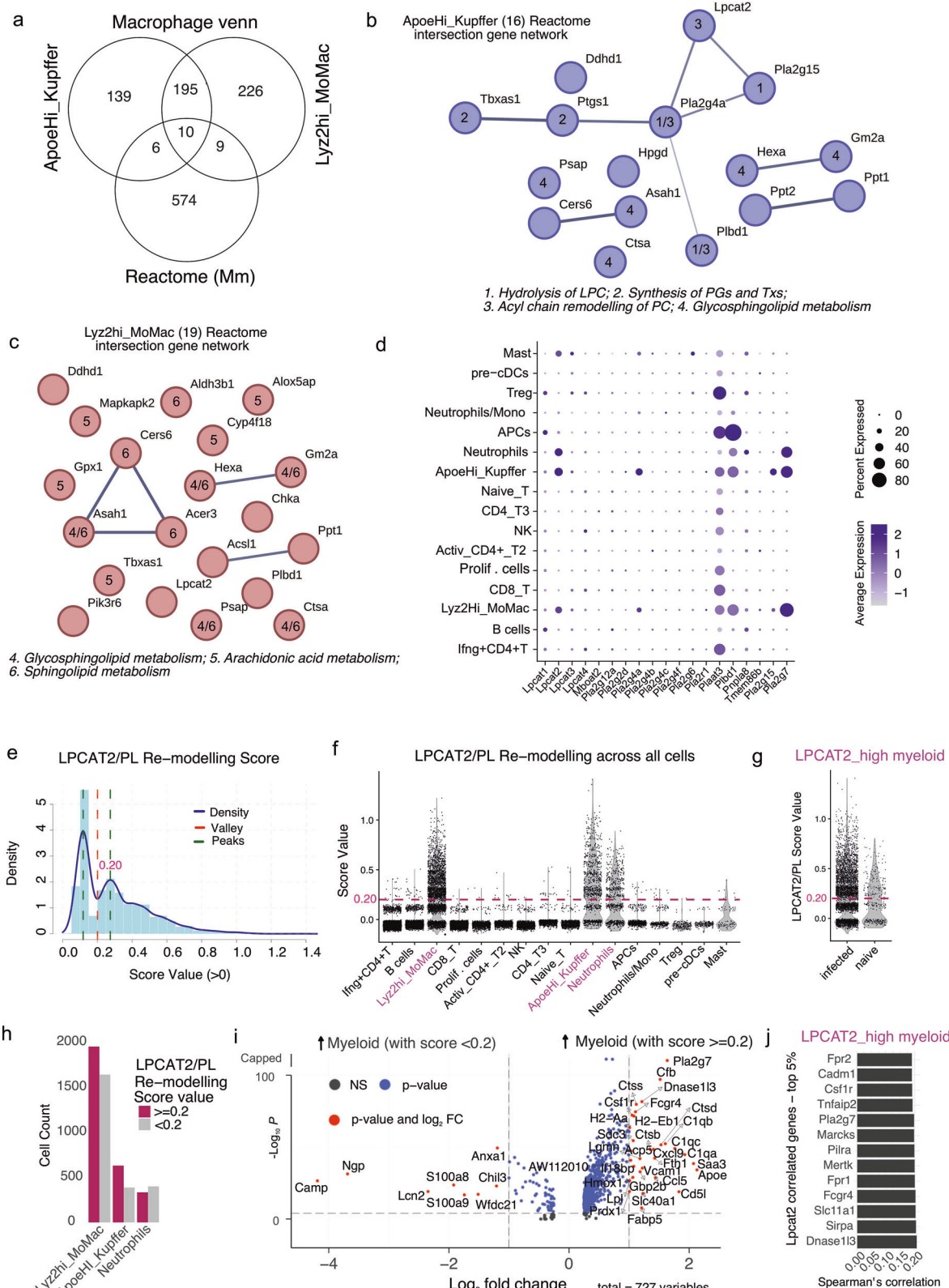

**Fig. 3 | Lands cycle enzyme LPCAT2 expression in resident and recruited macrophages. a–c** Venn diagram showing the intersection of Reactome Lipid metabolism genes for Mus musculus with top genes expressed by resident Kupffer and monocyte-derived macrophages (MDMs) (**a**). Genes identified in a are presented as a network of interactions as on STRINGDB and numbered by their role in various reactome pathways as indicated for Kupffer cells (**b**) and MDMs (**c**). **d** Dot plot showing the expression of Lands Cycle related genes across single-cell populations.

**e–g** Histogram showing the spread of module score LPCAT2/PL remodeling score (*Lpcat2*, *Pla2g4*, *Pla2g7*) for all cells (**e**). Score in e is plotted for each single-cell population (**f**) and for naïve versus infected mice (**g**). **h** Barplot showing split between populations of Kupffer cells, MDMs, and Neutrophils scoring high/low for LPCAT2/PL remodeling. **i** Volcano plot showing genes upregulated myeloid cells in h that score high or low. **j** Barplot showing Spearman's correlation between *Lpcat2* and other genes in high-scoring myeloid cells as in (**i**).

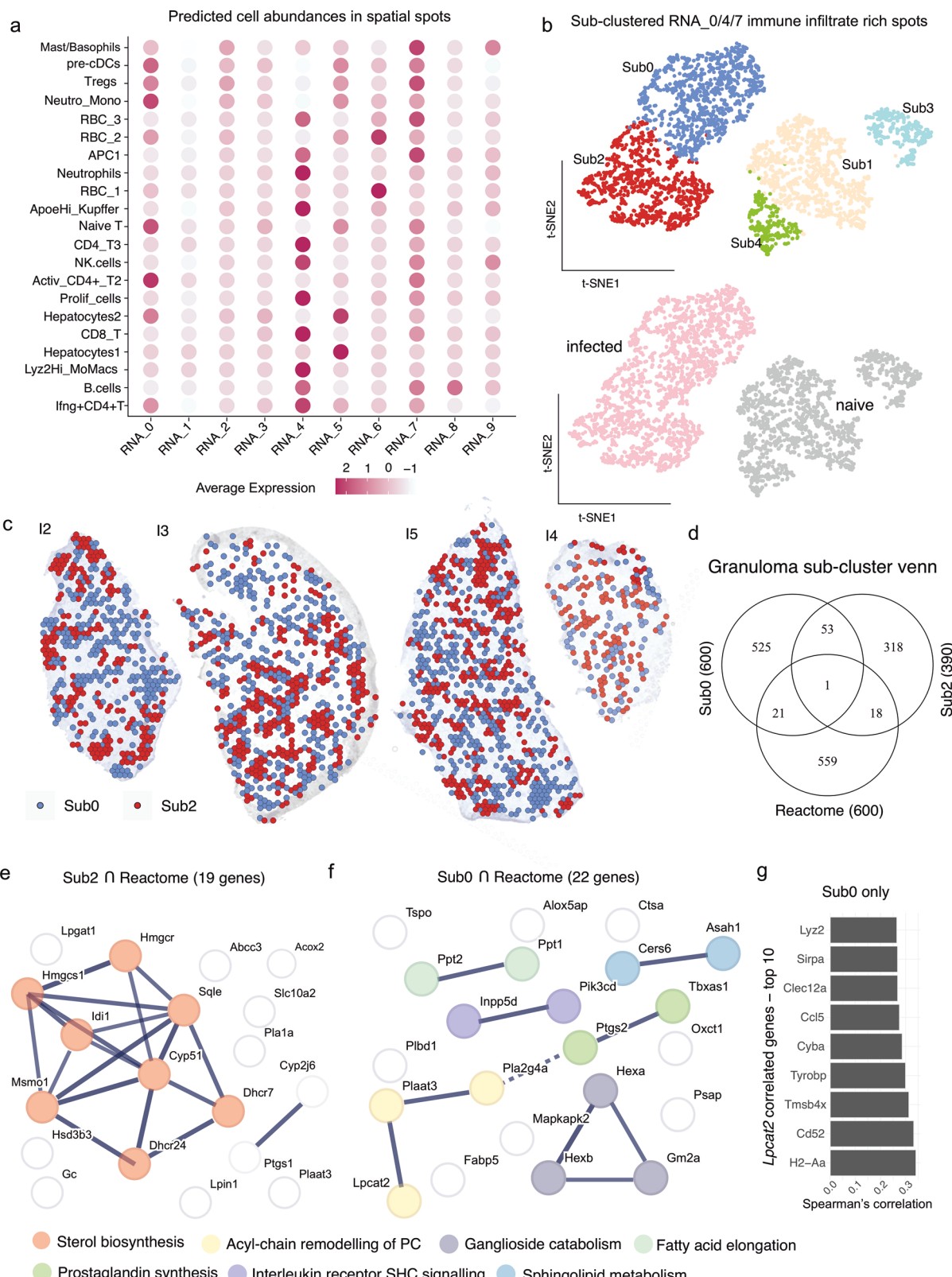

**Fig. 4 | Lipid metabolism related genes in granulomatous clusters. a** Dot plot showing predicted cell abundances in RNA_clusters identified in Fig. 1. **b** RNA_0/4/ 7 re-clustered to identify sub-clusters Sub0-4 (top) associated with infected or naïve mice (bottom). **c** Spatial representation of Sub0 and Sub2 on infected tissue (*n* = 4 mice). **d** Venn diagram showing the intersection of Reactome Lipid metabolism genes with top genes expressed by Sub0 and Sub2. E,f, STRINGDB network of intersecting genes for Sub2 (**e**) and Sub0 (**f**), colored by MCL clustering and function. Edge thickness represents interaction confidence and dotted lines indicate cluster boundaries. **g** Spearman's correlation between *Lpcat2* and genes in Sub0.

(Fig. 4b). Overlaying Sub0 and Sub2 on infected tissue demonstrated a spatial distribution supporting their assignment to granulomas (Fig. 4c). From our lipidomics dataset, Sub0 and Sub2 differentially contained major lipid classes including phosphatidylglycerol (PG), phosphatidylserine (PS), phosphatidylethanolamine (PE), sphingomyelin (SM), phosphatidylcholine (PC), ceramide phosphate (CerP), carnitine (CAR), and inositol-containing lipids such as phosphatidylinositol (PI) inositolphophoceramide (IPC). Ether-linked species were identified in both PC (PC(O-)) and PE (PE(O-)) classes, along with diacylglycerols (DG), diacylglyceryl trimethyl-lalanine (DGTA), triacylglycerol (TG), and cholesteryl ester (CE). Several of these lipids contained arachidonyl acid (20:4)-such as CE(20:4;O2) (Supplementary Fig. 6) and PE(O-18:0_20:4), LPE(20:4), PE(20:0_20:4), LPC(20:4), PS(18:0_20:4) in the top lipid species (Supplementary data 3) discriminating Sub0 and Sub2 from subclusters found in naïve mice. (Supplementary Fig. 6).

Comparing spatial gene expression profiles in Sub0 and Sub2 with Reactome Metabolism of Lipids (Fig. 4d) identified that sterol biosynthesis pathway was enriched in Sub2 (Fig. 4e), whereas Sub0 suggested enrichment of acyl-chain remodeling of PC. Of note, PG synthesis enzymes such as *Lpcat2*, *Plaat3*, and *Pla2g4a* observed in myeloid cells (Fig. 3d) were associated with Sub0 (Fig. 4f). Sub0 also showed enrichment for Sphingolipid metabolism (Fig. 4f). Within Sub0, *Lpcat2* correlated with *H2-Aa*, *CD52*, *Sirpa* amongst others (Fig. 4g).

## Acyl-chain remodeling in granulomas

We next compared the *Lpcat2, Pla2g4a, Plaat3, Pla2g15* and *Pla2g7* signature (Fig. 5a) and predicted cellular abundances (Fig. 5b) within Sub0 and Sub2. The identification of *Lpcat3* within Sub0 was consistent with the predicted presence of hepatocytes1[28] and may reflect some inclusion of hepatocytes in spots at the periphery of granulomas. *Lpcat1* was expressed in Sub0 spots and may reflect lymphocytes composition (Fig. 5b). ApoeHi_-Kupffer cells had similar predicted abundance between these subclusters, whereas Lyz2Hi_MoMac were less abundant in Sub2 (Fig. 5c). We further identified differentially expressed genes discriminating these two sub-clusters (Fig. 5d). To determine whether Sub0 and Sub2 also contained Lands Cycle products and / or substrates for LPCATs, we examined phosphatidylcholines (PCs) identified by MSI. Sub0 showed prominent abundance of PC(30:0), PC(32:0), PC(33:0) PC(34:3), PC(40:8), as well as of all detected lysoPCs LPC(16:0), LPC(17:0), LPC(18:0), LPC(18:2) and LPC(20:4) (Fig. 5e). Notably, LysoPCs showed spatial hotspots (Fig. 5f) and with saturated LPC(18:0) and LPC(16:0) intensities being highly correlated (Fig. 5g). Collectively, these integrated transcriptomic and lipidomic data identify altered expression of Lands cycle enzymes and their associated substrates (and products) within granulomas and implicate ApoeHi_-Kupffer cells and Lyz2Hi_MoMacs as the dominant cell types associated with this pathway.

## Expression of LPCAT2 in granulomas

To validate these findings at the level of protein expression, we stained liver tissue for LPCAT2 using two different panels (Supplementary Fig. 7). In keeping with the mRNA data, LPCAT2 was localized to granulomas in infected mice (Fig. 6a, b and Supplementary Fig. 7) and was shown to co-localize with SIRPα and NOS2 (Fig. 6a). We note that not all granulomas express NOS2 and that a visually higher LPCAT2 expression may associate with NOS2 expression (Fig. 6a and Supplementary Fig. 7). To potentially assign LPCAT2 expression to ApoeHi_Kupffer cells, we co-stained for LPCAT2 and ApoE (Fig. 6b). Based on dual expression of LPCAT2 and ApoE in manually demarcated and segmented granulomas (Fig. 6c), we compared the distribution of cell types based on ApoE and LPCAT2 expression using mean intensity per segmented cell (Fig. 6d). Amongst ApoE+ Kupffer cells, ~55% express LPCAT2 (Fig. 6e), with ApoE-LPCAT2+ cells likely reflecting LPCAT2 expressing Lyz2hi_MoMacs or neutrophils (Fig. 3f). LPCAT2+ Kupffer cells were found in 61% (range 0-100) of all granulomas examined (Fig. 6f). Only a weak correlation was observed between parasite abundance per granuloma (determined by OPB

staining) and the number of LPCAT2+ cells (Fig. 6g), suggesting LPCAT2 expression is driven by local macrophage activation and is largely independent of macrophage infection per se.

## LPCAT2 is associated with macrophage activation in vivo

We used ex vivo cell sorting followed by proteomics, to further analyze the nature of LPCAT2+ macrophages within the granuloma. As LPCAT2 is expressed intracellularly, we used SIRPA / CD172 as a surrogate cell surface marker, given that the *Sirpa* transcript and protein were correlated with granulomas (Figs. 4g and 6a) and macrophages with a high LPCAT2/PL remodeling score (Fig. 3j). We sorted gated CD3-CD11b+F4/80+ cells as CD172hi (68%) and CD172lo (24%) (Fig. 7a and Supplementary Fig. 8) and applied data independent acquisition proteomics to each population. As expected, sorted CD172hi cells had significantly higher abundance of CD172 and LPCAT2 compared to CD172lo cells but not ApoE, CD11b, F4/80, LPCAT1, LPCAT3 or LPCAT4 (Fig. 7b). The lack of differential ApoE expression suggested that each population of CD11b+F4/80+ sorted cells contained similar numbers of Kupffer cells (Fig. 7b). Next, we compared CD172hi and CD172lo cells by differential protein expression analysis. The top 5 proteins (by fold change) found in CD172hi cells were TRAPPC14, WWP2, TMEM41B, CCDC127, and TMEM260. CD172lo cells were enriched for MCPT8, PVALB, CALM3, SDR39U1 and KRT23 (Fig. 7d and Supplementary data 4).

GSEA analysis of all proteins upregulated ($\log_2$FC > 2; FDR 5%) in the CD172hi cells identified mitochondrial translation and TNFR1-mediated signaling (known to maintain inflammatory programming[29]) as enriched pathways (Supplementary data 5). CD86 protein was more abundant in CD172hi cells, whereas ARG1 was more abundant in CD172lo cells, suggesting differing states of activation between macrophages expressing physiologically higher or lower levels of LPCAT2 (Fig. 7c). Finally, we looked at proteins intersecting the Reactome Metabolism of Lipids. ~5% of proteins from CD172hi cells intersected this pathway (Fig. 7d) with pathway analysis indicating increased acylglycerol biosynthesis, and lysophospholipid acyltransferase activity, inositol phosphate metabolism, phosphatidylinositol 3-kinase complex activity, steroid biosynthesis, PPAR regulation, sphingolipid metabolism, and vesicle trafficking (Fig. 7e).

Collectively, this multimodal data indicates selective lipid-mediated cell membrane remodeling in ApoeHi_Kupffer cells and Lyz2Hi_MoMac at the heart of the granulomatous response to *L. donovani* infection.

## Functional role of LPCAT2 macrophage activation

Macrophage activation is critical for control of *L. donovani* infection and for inflammation resolution, with NOS2 playing a major role as a leishmani-cidal effector mechanism. We therefore sought to further characterize the relationship between LPCAT2 expression and NOS2 and the functional role of LPCAT2 in NOS2 regulation. Subpopulations of liver myeloid cells were first identified by flow cytometry (Supplementary Fig. 9a for gating strategies). Compared to naive mice, infected mice had higher LPCAT2 expression within a bulk CD11b+F4/80+ population (Fig. 8a). We then identified four subpopulations of CD11b+Ly6G- cells inflammatory monocytes, F4/80+Ly6c_high cells, transitioning Ly6c intermediate cells, and F4/80+Ly6c_low cells. Amongst these populations, F4/80+Ly6Chi cells increased most dramatically in infected mice (Fig. 8b, c). NOS2 expression was absent in naïve cells (Supplementary Fig. 9B) but observed in all four myeloid populations. Strikingly, NOS2 expression was largely restricted to cells expressing the highest levels of LPCAT2 (Fig. 8d, e).

To further explore the relationship of between LPCAT2 expression and pro-inflammatory markers, we stimulated bone marrow derived macrophages (BMDMs) from naïve C57BL/6 mice with LPS. As BMDMs showed heterogeneity in LPCAT2 expression, we gated these cells into LPCAT2_high and LPCAT2_low populations (Fig. 8f). LPCAT2_high cells expressed more TNF and NOS2 than LPCAT2_low cells (Fig. 8g). As IFNγ is highly expressed by T cells in *L. donovani* infection (Fig. 2b) and a strong driver of NOS2 production and macrophage activation, we next stimulated BMDM with LPS and IFNγ to determine the effects of LPCAT2 inhibition

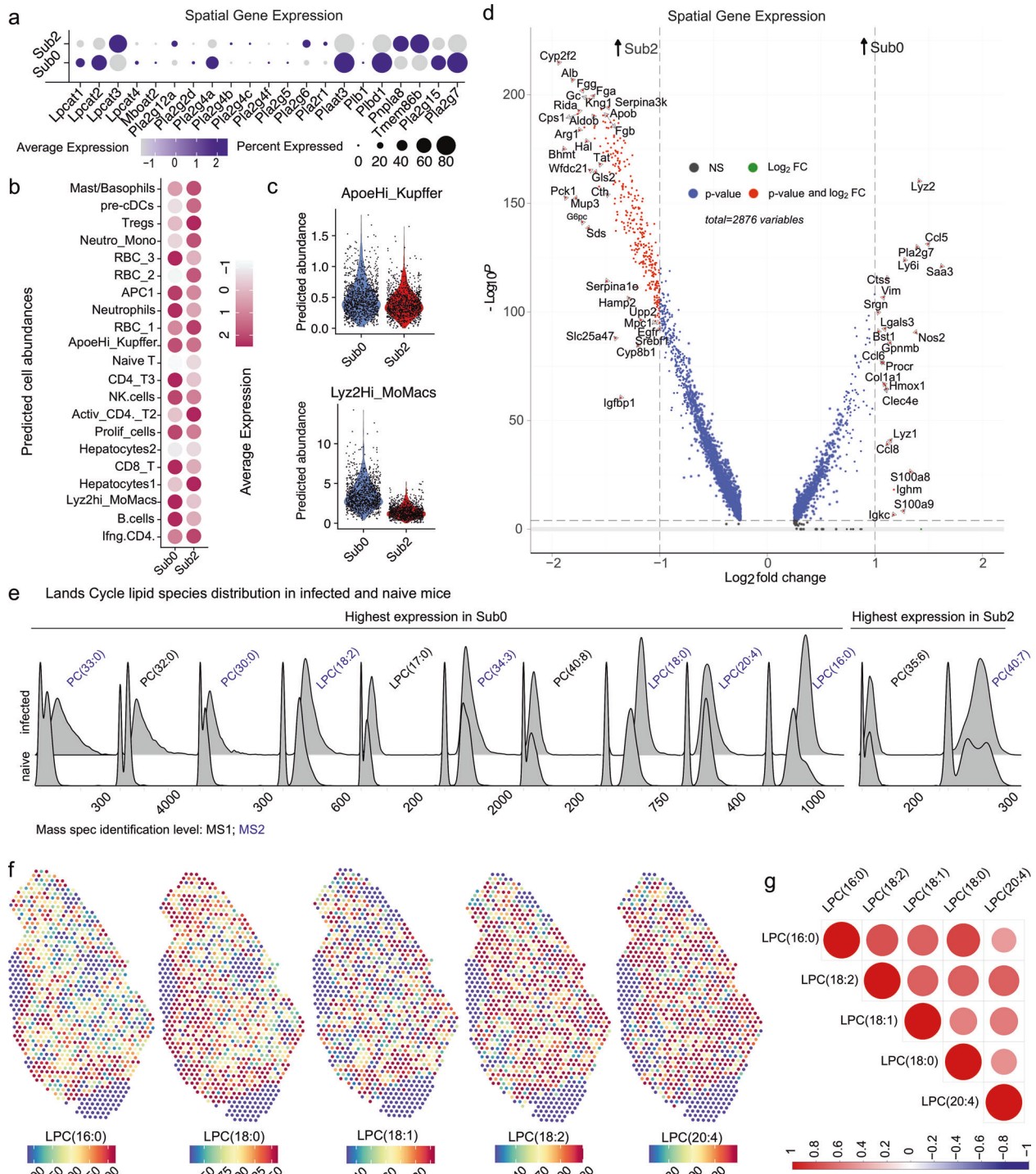

**Fig. 5 | Association of LysoPC species with immune granuloma spots. a** Dot plot showing gene expression between Sub0 and Sub2. **b** Dot plot showing predicted cell abundances in Sub0 and Sub2. **c** Violin plots showing predicted cell abundances for Kupffer cells and monocyte-derived macrophages between Sub0 and Sub2 where each dot represents an individual Visium spot. **d** Volcano plot showing genes upregulated in Sub0 versus Sub2. **e** Ridge plots showing distributions of LysoPCs (LPCs) and PCs with those identified with higher confidence (MS2; data-dependent lipid identification) indicated in blue across infected and naïve mice. **f** Spatial MSI intensity of LPCs. **g** Correlation plot (Spearman's) between LPCs.

on NOS2 production. In the presence of a specific pharmacological inhibitor, TSI-01[30], LPS/IFNγ-induced NOS2 expression was significantly reduced but not TNF (Fig. 8h). The impact of LPCAT2 inhibition on NOS2 expression was confirmed by the reduced levels release of nitrite an indirect measure of NO production (Fig. 8i). Collectively, these data indicate that macrophage activation leading to NOS2 expression and NO production involves processes governed by, or operating through, LPCAT2.

## Discussion

In this study, we demonstrate that LPCAT2-mediated phospholipid remodeling is a defining feature of hepatic granulomas during *Leishmania donovani* infection. Through an integrated analysis combining spatial transcriptomics, lipidomics, and proteomics, we report four major findings: First, LPCAT2 expression is upregulated in macrophages within granulomas and weakly correlates with parasite burden. Second, granulomas show

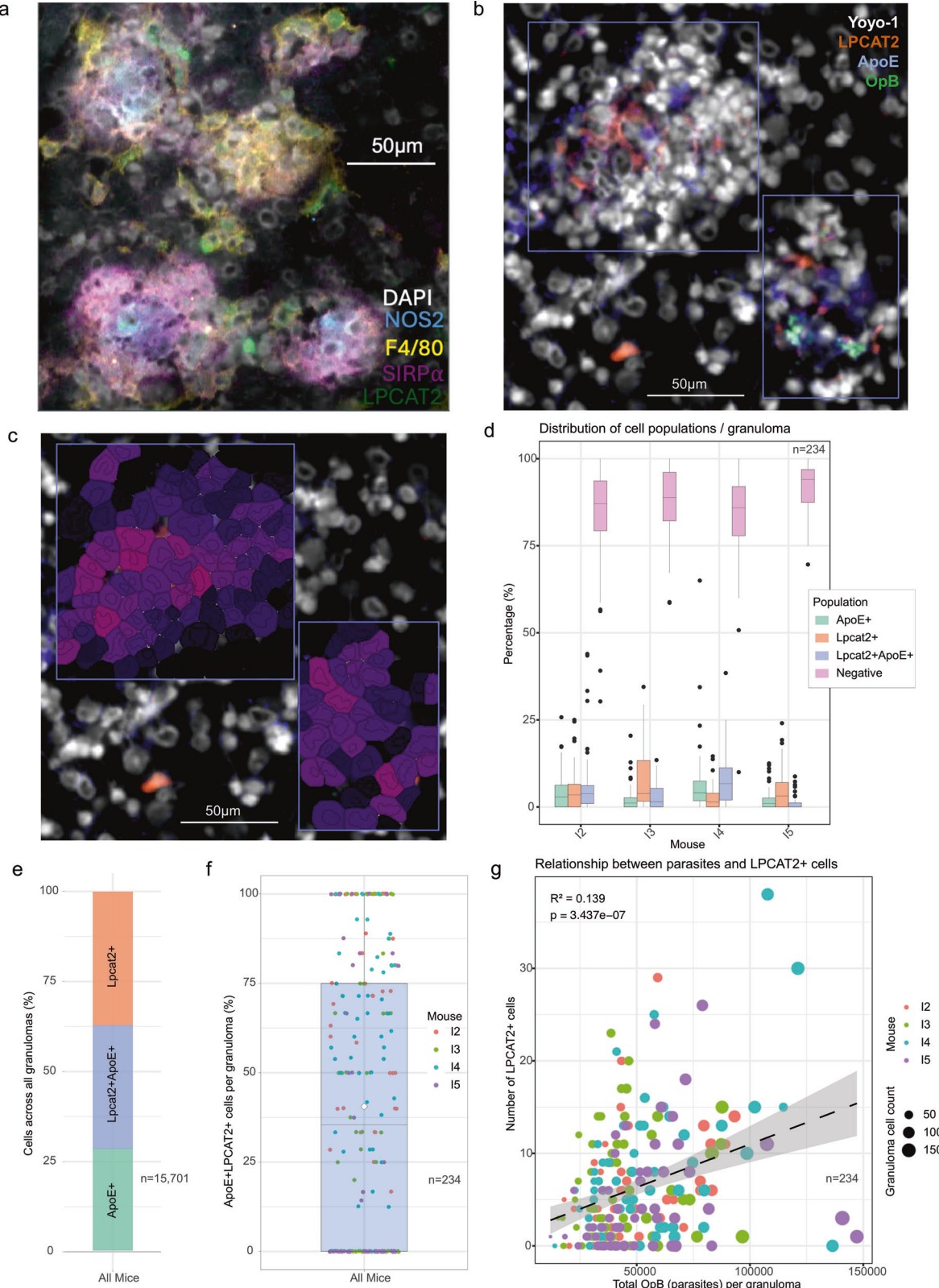

**Fig. 6 | LPCAT2 protein in granulomas. a** Representative immunohistochemistry image showing nuclei (DAPI) NOS2 (AF488), F4/80 (AF594) and SIRPα (AF647). **b** Same as a but nuclei (Yoyo-1), LPCAT2 (CF750), ApoE (AF594) and parasites (OpB; AF647). **c** Exemplar cell segmentation based on Yoyo-1 staining and colored on mean LPCAT2 intensity per pixel within cells (dark purple to light purple represents low to high intensity) from staining panel in (**b**). **d** Bar plots showing percentage of single positive (ApoE⁺ or Lpcat2⁺), double positive (Lpcat2⁺ApoE⁺)

or negative cells per granuloma. Total granulomas; n = 234. Box represent the interquartile range (IQR) with line representing median and whiskers show 1.5*IQR on either side. **e** Barplot showing percentage of myeloid populations based on LPCAT2 and ApoE expression for all cells across all measured granulomas. **f** Box and whiskers plot showing per granuloma occurrence of ApoE+ LPCAT2+ cells. **g** Scatter plot between LPCAT2⁺ (sum of Lpcat2⁺) and parasites (sum of OpB staining) per granuloma.

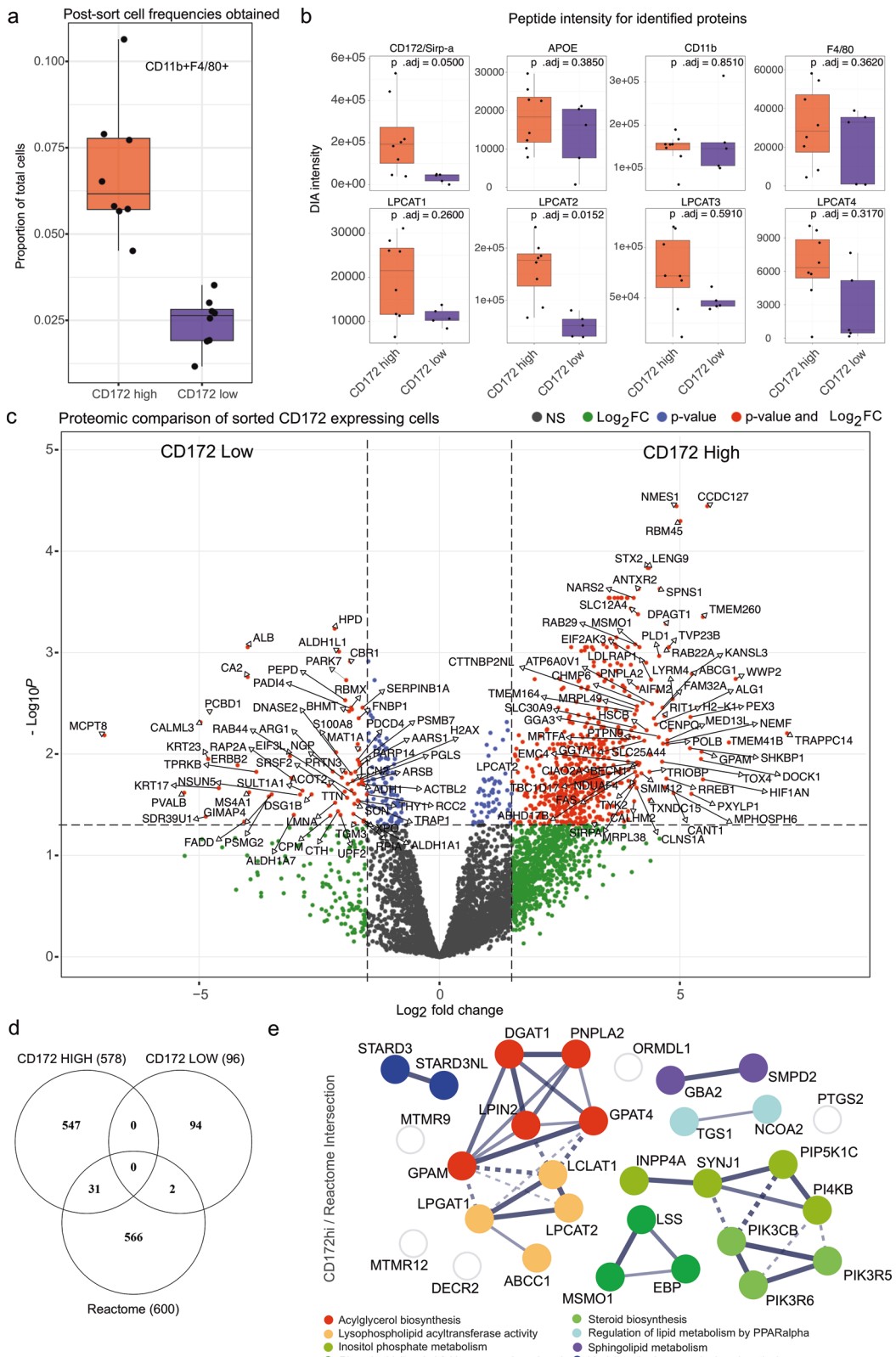

**Fig. 7 | Proteomic analysis of LPCAT2 high/low macrophages. a** Box and whiskers plot showing the proportion of CD172hi versus CD172lo macrophages. Sort gates described in Supplementary Fig. 8. **b** Proteomic peptide intensity between samples (*n* = 8 for CD172hi; *n* = 5 for CD172lo) for SIRP-α/CD172, APOE, CD11b, F4/80 and LPCAT1-4. **c** Volcano plot showing differentially expressed proteins between CD172hi versus CD172lo macrophages. **d** Venn diagram showing the intersection of

Reactome Lipid metabolism genes for *Mus musculus* with top proteins expressed by CD172 high and CD172 low macrophages. Panels **c** and **d** indicate FDR corrected p-values, see "Methods" *FACS Sorted Proteomics*. **e** Proteins identified in (**c**) and genes identified in (**d**) are presented as a network of interactions using STRINGDB and colored by MCL clustering. Edge thickness represents interaction confidence and dotted lines indicate cluster boundaries.

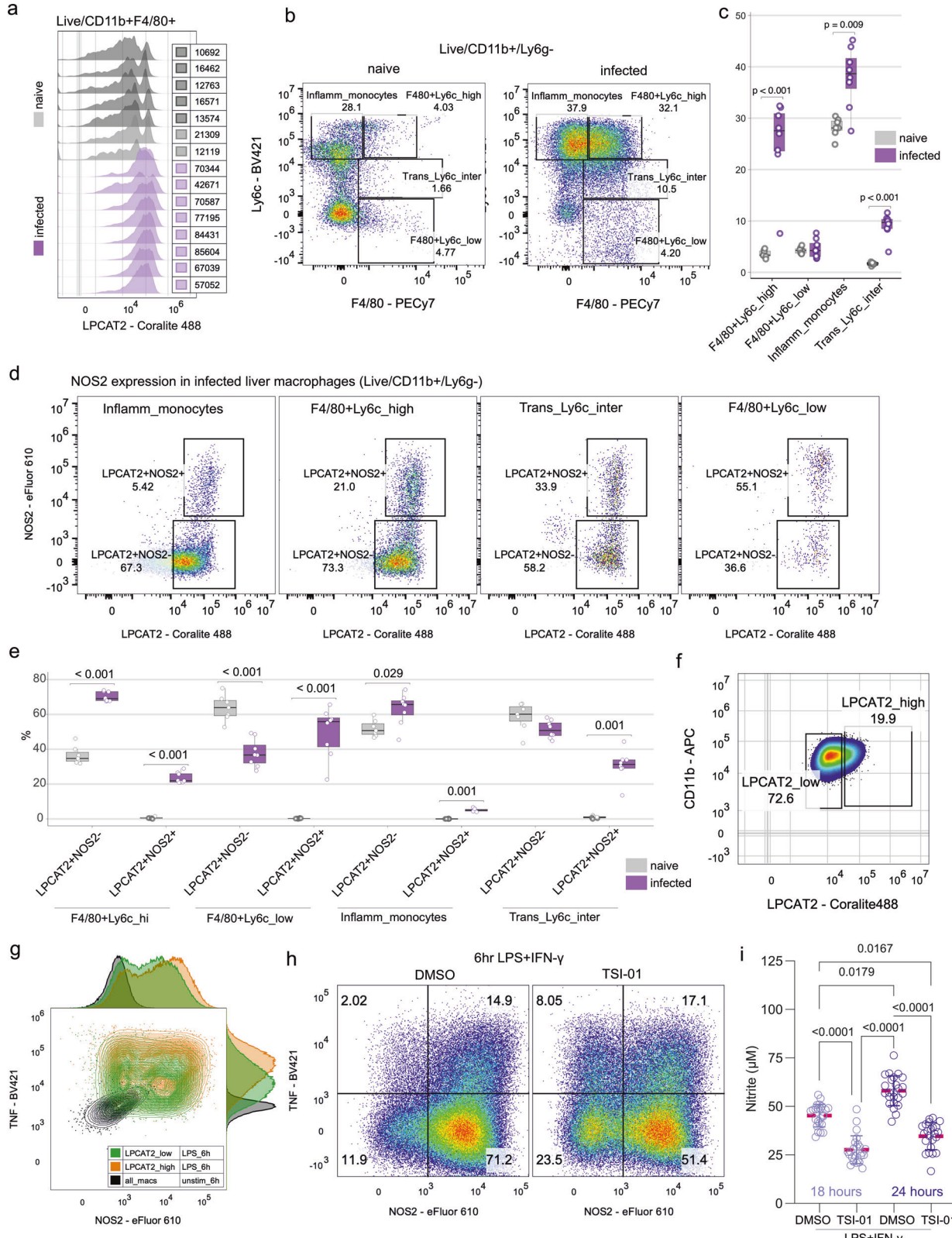

**Fig. 8 | Regulation of NOS2 expression by LPCAT2. a** Histogram overlay comparing LPCAT2 expression (Coralite 488) in CD11b⁺F4/80⁺ liver macrophages from naive (gray; *n* = 7) and infected (purple; *n* = 8) mice. Median fluorescence intensity (MFI) is indicated. CD11b⁺Ly6g⁻ subpopulations in naive and infected mice shown in a representative dot plot (**b**) and quantified as % per mouse (**c**). **d, e** Representative dot plot (**d**) and quantification of LPCAT2 and NOS2 expression in myeloid subsets as % per mouse. *P* values derived from Wilcoxon-rank sum test with Bonferroni correction and box and whiskers plot where box indicates interquartile range and median. **f** Bone marrow derived macrophages were gated as LPCAT2hi

and LPCAT2lo cells and **g** analysed from LPS-induced TNF and NOS2 expression. LPCAT2_high (orange) versus LPCAT2_low (green) are shown with adjacent overlaid histograms. Gray contours show unstimulated cells. **h** NOS2 and TNF expression following 6-h LPS + IFN-γ stimulation in the presence of vehicle (DMSO) or LPCAT2 inhibitor, TSI-01. **i** Nitrite production by LPS and IFNγ stimulated BMDMs at 18- and 24-h in the presence of vehicle or TSI-01. Data presented as mean ± SD; *P* values derived from Kruskal-Wallis multiple comparison statistic.

distinct accumulation of lysophosphatidylcholines (LPCs), the key substrates for LPCAT2, suggesting active phospholipid remodeling in these inflammatory microenvironments. Third, macrophages with high LPCAT2 expression display a pro-inflammatory phenotype characterized by their transcriptomic and proteomic signatures. Fourth, pharmacologic inhibition of LPCAT2 significantly reduces macrophage NOS2 expression and nitrite production.

In our analysis, we define three main populations of monocytes and macrophages in the liver of infected mice. ApoeHi_Kupffer cells have abundant mRNA for *Adgre1*[+] (F4/80), *Marco*, *Clec4f*, and multiple complement components and most likely represent liver resident embryonically derived Kupffer cells[31]. Of note, ApoE expression is associated with efflux of cellular cholesterol in cholesterol-loaded macrophages[32,33] and cholesterol depletion may favor parasite survival through indirect effects on antigen presentation[34] or direct effects on parasite survival[35]. Lyz2Hi_MoMac have abundant *Ccr2*[+] mRNA but little *Ly62c*, and given their increased representation in infected compared to naïve mice, these likely represent monocyte-derived macrophages, that may eventually transition to a Kupffer cell phenotype[31,36,37] These Lyz2Hi_MoMac however, have a mixed composition in terms of activation markers, expressing *Chil3* or Ym1 known to promote Th2 immunity[38] as well as *Tnf*, *Tlr2* and *Cd86*. This mixed phenotype is perhaps not surprising given the heterogeneous cytokine profile associated with hepatic *L. donovani* infection[39]. Whether this further reflects subtleties in microenvironment or activation-induced heterogeneity in macrophages[40] remains to be determined. Finally, we identified a smaller population of monocytes with abundant *Ly6c2* but low levels of *Ccr2* mRNA. Of these three populations, only ApoeHi_-Kupffer cells and Lyz2Hi_MoMac expressed LPCAT2.

Analysis of lipid metabolism in these two macrophage populations indicated that they were enriched for SM metabolism, arachidonic acid (AA) metabolism, and acyl-chain remodeling of PCs via *Lpcat2*. LPCAT2 has been shown translocate to membrane lipid rafts and promote inflammatory gene expression[41]. LPCAT2 also participates in the Lands Cycle that re-acylates LysoPCs capable of incorporating polyunsaturated fatty acids[22] such as arachidonic acid (AA) into the membrane that can lead to downstream processing of lipid mediators of inflammation[25]. By generating a LPCAT2/PL remodeling score (*Lpcat2, Pla2g4a, Pla2g15, Pla2g7*), we found both populations contained high scoring cells in comparable proportions. The LPCAT2/PL remodeling score also positively correlated with a pro-inflammatory score.

Integrating data across modalities, we showed that the substrates for LPCAT2, lysoPCs as identified by mass spectrometry imaging are associated to immune granulomas. Palmitoyl-lysoPC(16:0), lysoPC(17:0), stearoyl-lysoPC(18:0), linoleoyl-lysoPC(18:2) and arachidonoyl-lysoPC(20:4) were all detected. Palmitoyl and arachidonyl-lysoPCs are known to promote Cox-2 expression[42], a key enzyme for AA-led inflammation, and *Ptgs2* (Cox-2) and *Tbxas1*, involved in downstream AA metabolism, were upregulated in macrophages and localized to granulomas. These data are in accord with our previous observation of enrichment of AA-containing lipids in *Leishmania* granulomas[20].

We confirmed the expression of LPCAT2 by ex vivo proteomic analysis of macrophages, using SIRPA / CD172 as a surrogate for LPCAT2 expression. In line with in vitro findings of LPCAT2 having a pro-inflammatory effect on macrophages[41] we found that CD172[hi] cells were enriched for pro-inflammatory pathways. However, macrophage phenotype with respect to lipid composition is clearly complex and likely involves elaborate lipid remodeling by lysophospholipid acyltransferases not limited to LPCAT2. For example, multiple ether-linked PCs that have been linked to be sensitive for ferroptosis[43] such as PC(O-36:3), PC(O-34:2) were also found in granulomas. Sorted CD172[hi] macrophages also expressed PEX3, a peroxisomal biogenesis factor essential for ether-linked PC synthesis[44].

The spatial restriction of LPCAT2 expression to areas of granulomatous inflammation suggests that local microenvironmental cues are the primary drivers of LPCAT2 expression. Several factors likely contribute to this localized activation signature. First, the granuloma cytokine milieu, particularly IFNγ and TNFα as evidenced by our transcriptomic analysis,

creates a pro-inflammatory microenvironment that favors LPCAT2 upregulation. Second, *Leishmania* infection rapidly increases AA metabolism[45], potentially creating positive feedback loops that sustain LPCAT2 expression for continued membrane remodeling. Third, the fact that both tissue-resident Kupffer cells and recruited monocyte-derived macrophages express LPCAT2 suggests this represents a response to granulomatous inflammation rather than a cell-type-specific programming. Interestingly, we note inflammatory monocytes in infected mice also express high expression of LPCAT2 and NOS2 suggesting that LPCAT2-expressing migratory monocytes may also have a more systemic role in regulating immunity and inflammation during *L. donovani* infection.

We demonstrated a direct causal relationship between LPCAT2 and macrophage activation through pharmacological inhibition of LPCAT2 in BMDM in vitro. BMDM treated with the LPCAT2 inhibitor TSI-01 expressed reduced levels of NOS2 and produced less NO, suggesting a therapeutic potential for LPCAT2 inhibition in conditions driven by overt inflammation and/or where excess NO production is pathologic. Nevertheless, it remains to be determined how inhibition of LPCAT2 might play out in vivo, where there exists a delicate balance between the pro- (PAF-mediated) and anti-(prostaglandin-mediated) inflammatory activities, the requirements for parasite control and the ultimate need for resolution of inflammation. To address these and other questions, Cre-driven cell-specific LPCAT2 knockout models will likely be crucial.

Finally, our data have implications beyond leishmaniasis for the tissue- and context-specific regulation of lipid metabolisms by LPCAT enzymes. In lung homeostasis, acyl chain remodeling of phospholipids essential for pulmonary function occurs through LPCAT1[46], and this pathway is enriched in lung disease[47]. In liver homeostasis, LPCAT3 is important for lipid homeostasis[28], and in our study was upregulated in infected mice in spots rich in hepatocytes. However, within immune granulomas LPCAT2 is highly expressed in both resident and recruited macrophages. LPCATs and their substrates LPCs have high therapeutic potential, being implicated in conditions from pain[48] to cancer metastasis[49]. They also function as lyso-platelet activating factor (PAF) acyltransferases, synthesizing PAF in a range of proinflammatory diseases, and selective LPCAT2 inhibitors have recently been identified[30] opening the door for future therapeutic manipulation and further mechanistic studies in pre-clinical models.

Our study has some limitations. Whilst our spatial resolution is 55μm, spots have a center-to-center distance of 100 μm resulting in voids where measurement does not occur. Similarly, some spots on the periphery of granulomas may include parenchymal cells. These issues are partly mitigated, however, by the sampling of large numbers of granulomas and their lack of a defined shape / compartmentalization. The marriage between single cell and spatial information is also largely probabilistic and not exact, but we have used orthogonal approaches to cross validate key findings. Finally, although we observed a weak but significant correlation between LPCAT2 expression and granuloma parasite load, it was not possible to definitively link intracellular parasitism with the changes observed. Our data can be further probed when technical challenges of reliably identifying parasite transcripts are overcome. Although parasites were observed in some granulomas, thin tissue sections do not sample the entire granuloma volume and may underestimate parasite load. Given that parasites are largely found in embryonically derived Kupffer cells[37] but we found that both ApoeHi_Kupffer cells and Lyz2Hi_MoMac regulate LPCAT2, it is likely that many of the signals governing macrophage activation state and LPCAT2 expression are cell-extrinsic and derived from other leukocytes in the granuloma microenvironment.

In conclusion, our study provides a comprehensive map of the immunometabolic landscape within *L. donovani*-induced granulomas. The identification of LPCAT2 as a key player in the macrophage response during granulomatous inflammation brings forth new avenues for understanding and potentially modulating the immune response in leishmaniasis and possibly other granulomatous diseases. Future studies should focus on elucidating the functional consequences of lipid remodeling in granulomas and exploring the therapeutic potential of targeting these pathways.

## Materials & methods

### Ethics statement

All animal studies were performed under UK Home Office Licence (PP0326977) and were approved by the Animal Welfare and Ethical Review Board of the University of York. We have complied with all relevant ethical regulations for animal use.

### Animals and infection model

Adult female C57BL/6 J mice were maintained and bred at the University of York, UK. The mice were housed in ventilated cages under specified pathogen-free conditions with access to food and water ad libitum. Identification of the mouse colony was confirmed through genetic profiling of microsatellite markers, verifying C57BL/6 J genetic background with minor variations at 3 markers. Healthy (6–8 week old females) were either infected on day 0 intravenously with $3 \times 10^7$ amastigotes of the Ethiopian LV9 strain of *Leishmania donovani* to establish visceral infection or left naïve. At 28 days post-infection, mice were euthanized by $CO_2$ inhalation followed by cervical dislocation to model peak granulomatous inflammation in the liver. Livers were harvested after euthanization and the major lobe cut and snap-frozen in liquid nitrogen while bathed in isopentane and then kept in -80 degrees until used. The remaining liver was harvested in 2% fetal calf serum (FCS) in 1x PBS and immediately processed into a single cell-suspension. Age-matched naïve female mice from the same colony were used as baseline controls for liver tissue. Individual mice were used as experimental units unless stated in the figure legends. Aged-matched mice were randomly (coin-toss) allocated to groups, and no blinding or exclusion was employed due to clear experimental conditions (infected vs. naive mice). Specific a priori effect size calculation for the multi-modal analysis was not performed. A total of 40 mice were used across 4 independent infection experiments: Spatial transcriptomics and mass-spectrometry based lipidomics ($n = 8$; 4 infected and 4 naïve), single-cell RNA-seq ($n = 12$; 6 infected (3 matched mice from spatial transcriptomics and 3 from an independent experiment) mice plus 2 pools of 3 naïve mice each), flow-sorted proteomics ($n = 8$ infected mice, from which CD172[hi] macrophages [$n = 8$ samples] and CD172[lo] macrophages [$n = 5$ samples] were obtained), and flow cytometry characterization of LPCAT2 and NOS2 expression in myeloid populations ($n = 7$ naïve and $n = 8$ infected). From flow sorted samples, due to low cell recovery post-sort in the CD172[lo] fraction, led to three unmatched CD172[hi] animals.

### Liver dissociation and single-cell suspension

Liver tissue in 2%FCS in 1xPBS was sliced coarsely using a scalpel and then kept at 37 degrees in an incubator with 5% $CO_2$ in HBSS with collagenase and DNase for 20 min. Ice cold FCS containing RPMI was used to stop the enzymatic digestion by collagenase. Tissue was then passed through a 100µm cell strainer by mashing the tissue with the plunger of a 5 ml syringe. Upon completion, cells were suspended in ACK lysing buffer for 5 min. Cells were washed and then were centrifuged for 15 min without breaks in 33% Percoll to separate the hepatocytes. Hepatocytes formed a brown ring on top which was then discarded by decanting. The remaining cell population was washed twice and then re-suspended in PBS containing 0.05% BSA as per 10x instructions. Cells were either used for single-cell RNA seq or flow cytometry.

### Cryo-sectioning and histopathology

Frozen sections were cut in a Leica cryostat while maintaining them on a pedestal of OCT without allowing OCT to touch the tissue itself. 7 µm sections were cut onto PEN membrane slides, 10x Visium spatial transcriptomics gene expression slides and onto superfrost glass slides. The order of the sections was always, super frost 1, PEN membrane 1, Visium 1, Pen Membrane 2, superfrost 2 for each mouse. PEN membrane 1 and 2 were used for mass spectrometry-based imaging in positive and negative ionization mode measurements. The Visium slide was used for spatial transcriptomics.

### Mass spectrometry imaging of lipids and lipid identification experiments

All PEN microdissection membrane slides were dried for 15 min under vacuum and fiducial markers (Tipp-Ex, BIC, France) were applied to the slides before matrix application. The norharmane matrix solution at concentration of 7 mg/ml was prepared in 2:1 chloroform: methanol (v:v). An automated TM sprayer (HTX Technologies, Chapel Hill, NC, U.S.A.) was used for matrix application. In short, 12 layers were sprayed homogeneously at a fixed flow rate of 0.12 ml/min with the spray nozzle temperature at 30 °C combined with drying time between every layer for 30 s. After matrix application, lipidomics mass spectrometry imaging experiments were performed using a timsTOF fleX (Bruker Daltonik, Bremen, Germany) at the mass range of *m/z* 300–1800 with laser frequency of 5000 Hz and 50 laser shots accumulated per pixel in both negative and positive ionization modes. A total of 8 liver sections from 8 mice were measured with a pixel size of 20*20 µm2 for each polarity.

For the lipid identification, data-dependent analysis (DDA) imaging experiments were performed on 2 representative mouse liver sections (one naïve mouse and one infected mouse) using an LTQ Orbitrap Elite mass spectrometer (Thermo Fisher Scientific, Bremen, Germany) as previously described by Ellis et al. In short, a 240,000-mass resolution full-scan FTMS (Orbitrap) analysis was conducted in the mass range between *m/z* 300–1800 combined with stage step size of 25 µm (horizontal) and 50 µm (vertical). Subsequently, DDA IT-MS/MS (ion trap) scans were performed at adjacent 25-µm positions.

Automated lipid identification was performed using LipoStarMSI[50] (edition 2023) incorporated with the publically available Lipid Maps database. Precursor *m/z* tolerance was 0.00 Da ± 3.00 ppm and MS/MS *m/z* tolerance was 0.25 Da ± 0.00 ppm. The considered ion adduct types are $[M + H]^+$, $[M+Na]^+$, and $[M + K]^+$ for the positive ion mode and $[M-H]^-$ for the negative ion mode.

### Histological characterization & annotations for MSI

After MALDI-MSI measurement, the matrix was removed with 100% ethanol prior to hematoxylin staining. All slides were firstly rinsed in a graded ethanol series (100%, 96%, 70%, 2 min each), followed by immersing in distilled water for 2 min. Subsequently, the sections were stained with 0.1% Gill's Hematoxylin (Merck, Darmstadt, Germany) for 3 min, followed by rinsing in running tap water for 3 min and dehydrated in a graded ethanol series (70%, 2*96%, 2*100%, 2 min each). Finally, all slides were washed in 100% xylene for 2 min and not covered with cover slip. After drying overnight in a desiccator, the hematoxylin-stained slides were scanned using Leica Aperio CS2 slide scanner at 20x magnification (Leica Microsystems, Nussloch, Germany). All scanned high-resolution hematoxylin-stained images were coregistered in FlexImaging v5.0 (Bruker Daltonik GmbH, Bremen, Germany) to the MSI data using the previously applied fiducial markers.

The acquired MSI data and co-registered hematoxylin-stained images were imported into the mass spectrometry imaging software SCiLS Lab (version 2023b) for both polarity experiments separately without baseline removal and automatic resampling. Pixels of the negative and positive polarity datasets were normalized to their root-mean-square or total-ion-count values, respectively. Peak picking was done on the average spectrum of every dataset using mMass and the following parameters: S/N > 5, relative intensity threshold >0.2%, picking height 90%, and de-isotoping with mass and intensity tolerances of *m/z* 0.1 and 70%, respectively. Peak lists were re-imported with peak area as interval processing mode and a fixed interval width of 6 mDa and 10 mDa for the positive and negative polarity data, respectively, into SCiLS Lab.

### Spatial transcriptomics

Sections from uninfected and *L. donovani* infected mice ($n = 4$ per group, 8 total) attached to 10x Genomics Visium slides were processed using the Visium Spatial Gene Expression workflow as per manufacturer's instructions for fresh frozen tissue. Sections were fixed in pre-chilled methanol and

then stained in hematoxylin for 5 min, bluing buffer for 1 min and finally stained in eosin Y for 30 s. Coverslips were added using glycerol containing RNase inhibitor and the slides imaged using a Zeiss axioscan slidescanner. Coverslip was removed in MQ water and then the slides were placed in a gene expression cassette. H&E stained tissue sections were then permeabilized by proteinase K then reverse transcribed to synthesize cDNA directly on the slide. Second strand cDNA synthesis was performed for 25 min followed by cDNA denaturation for 15 min. The optimal number of amplification cycles was determined by qPCR then cDNA was amplified. Amplified cDNA was cleaned up with SPRIselect beads and quantified.

Fragmentation, end repair, A-tailing and adaptor ligation were performed to construct sequencing libraries. Post-ligation cleanup, sample index PCR and double-sided SPRIsize selection were conducted. Final libraries were quality controlled on an Agilent Bioanalyzer. cDNA libraries were constructed per kit instructions. Quantified libraries were sequenced on an Illumina NovaSeq 6000 to generate spatially barcoded, transcriptomic sequencing data aligned to H&E reference images Raw sequencing data was aligned to the mouse mm10 genome using 10x spaceranger (1.3.0) software. H&E images were aligned to fiducial markers on each slide using the Loupe browser. Tissue regions were manually selected in Loupe and coordinate JSON files created. These JSON files were input into spaceranger count() alongside sequencing data to generate spatially-resolved gene counts. Raw counts were loaded as a .h5 file in Seurat[51] for analysis.

### Image registration

To assist image registration between MSI and spatial transcriptomics (Visium) data, segmentation images were first created for the MSI data of infected mice in SCiLS lab using bi-secting k-means with correlation distance. Subsequently, the peak picked and normalized MSI data, its co-registered H&E image (H&E$_{MSI}$), and existing segmentation images were read-out from SCiLS Lab directly to R (version 4.1.0) using the API package "SCiLSLabClient". From there the data was transferred to MATLAB (version 2018a) via a .mat file using the R package "R.matlab". In MATLAB naïve mouse MSI datasets were segmented using k-means with $k = 3$. Finally, the Visium data and its associated H&E images (H&E$_{Visium}$) were also imported into MATLAB where the registration between MSI and Visium has been performed in a 2-step approach: First, a manual, coarse control point-based affine registration has been performed between H&E$_{MSI}$ and H&E$_{Visium}$. Since the MSI data is also registered with a certain error to its H&E$_{MSI}$, the MSI data was in a second step directly registered to the H&E$_{Visium}$ using the MSI segmentation image which shows morphological structures that can be found back in H&E$_{Visium}$. This step can be considered a refinement of the first registration and was also done manually using control points with affine or with piece-wise linear transformation, depending on the individual sample pairs' needs. This process was repeated for all individual datasets and resulted in the assignment of approximately 4–6 MSI pixels to the 55 μm-sized Visium spots. Pixels failing co-registration between MSI and Visium were excluded from downstream analysis, but this did not lead to animals being excluded from the study.

### Immunohistochemistry

Frozen liver sections (7μm) from naive and infected mice were cut using a Leica cryostat, placed onto superfrost slides. Sections were fixed in ice-cold acetone for 10 min followed by two washes (5 min each, with agitation) in wash buffer (PBS with 0.05% BSA, w/v). Hydrophobic barriers were then applied around each tissue section. Sections were blocked for 30 min at room temperature (RT) using dilution buffer comprising 5% donkey serum diluted in wash buffer supplemented with 0.1% Triton X-100. Primary antibodies targeting OpB (in-house; sheep IgG, 10 μg/ml) and LPCAT2 (Invitrogen PA5-101406; 1:500) were applied simultaneously in blocking buffer for 1 h at RT. Secondary antibodies donkey anti-sheep IgG AF647 (1:750 dilution) and donkey anti-rabbit IgG CF750 (1:500 dilution) were subsequently applied for 1 h at RT, prepared in wash buffer. Next, directly conjugated anti-ApoE antibody (rabbit monoclonal clone EPR19392, AF594 conjugate, Abcam ab310350, 1:500 dilution) and nuclear stain Yoyo-

1 (0.2 μM, 1:500 dilution from 100 μM stock solution) were applied together in wash buffer for 1 h at RT. Alternatively, for another panel anti-LPCAT2 was used as primary with CF750 secondary (as above) but with the following directly conjugated antibodies—NOS2 (clone CXNFT, eBioscience, 53-5920-82, Alexa Fluor 488, 1:250), anti-mouse F4/80 (clone BM8, BioLegend, 123140, Alexa Fluor 594, 1:250), and anti-mouse SIRPα/CD172a (clone P84, BioLegend, 144027, Alexa Fluor 647, 1:200), followed by DAPI (1:5000) as counterstain for 5 min.

Finally, sections were mounted using Prolong Gold mounting media and incubated at 4 °C in the dark for 24 h prior to imaging. Scanned images were imported into QuPath (v0.5.0) and granulomas were manually annotated (g = 243 granulomas containing c = 15,701 cells across $n = 4$ infected mice) and then cells within annotated granulomas were segmented using Yoyo-1 to obtain single-cell quantification of staining intensity.

### Single-cell RNA sequencing

Single cells were resuspended at a concentration of approximately 1,000 cells per microliter. Single cell RNA sequencing library preparation was then carried out using the Chromium Next GEM Single Cell 3' Kit v3.1 from 10X Genomics (Pleasanton, CA), following the manufacturer's CG000315 Rev A user guide. For library preparation, a target of 10,000 cells was captured per sample. Sequencing libraries were prepared as outlined in the 10X Genomics user guide and sequenced on the Illumina NovaSeq 6000 system (San Diego, CA) to achieve a minimum sequencing depth of approximately 20,000 reads per cell.

### Single-cell RNA data pre-processing

The raw FASTQ files generated from sequencing were aligned to the mouse reference genome (mm10) using the Cell Ranger software (version 6.1.0) from 10x Genomics. Cell Ranger Count was used to align reads and generate gene-barcode matrices for each library. The resulting .h5 files were imported into the R package Seurat (version 4.3.0) for quality control and downstream analysis. Low quality cells were filtered out by removing any cells with greater than 10% mitochondrial reads. The data was visualized using Seurat, plotting cells based on feature-feature relationships to identify cell populations and subset the data accordingly. Potential library construction artifacts were removed by dropping read counts for transcripts Gm42418 and AY036118. Data was normalized using SCTransform, implementing a gamma-Poisson generalized linear model (glmGamPoi)[52] to regress out variation due to mitochondrial reads, ribosomal genes, total RNA content, and unique feature counts. This normalized, filtered gene count matrix was used for subsequent analyses. Similar approaches for quality control of the spatial data were performed.

### Spatial data analysis

The positive and negative matrices per Visium pixel as generated from image co-registration resulted in an averaged matrix of MSI intensities [Vm x Mi] with species m/z measure in the positive and a second matrix [Vm x Mj] in the negative ionization mode Vm refers to 'm' spatial barcodes measured by the 10x Visium platform. The .h5 file from spatial transcriptomics also contained a matrix [Vm x Tk] which represented 1..k transcripts measured across 1..m barcoded spots from now referred to as the transcript matrix. At first, the matrices from positive and negative ionization measurements were combined i.e., a full join was used to obtain all Vn spots measured resulting in a matrix [[Vm x Mi] | [Vm x Mj]] or referred to as the MSI matrix [Vm x Mi+j].

Initially, each of the matrices were analysed separately for variable features and a principal components (PC) analysis was performed separately. The reduced PC space was then used as an input for finding clusters using Louvain clustering and as an input for t-SNE. Each Vm was then colored by its cluster identity and visualized in the 2-dimensional t-SNE space. When the PCs were deduced from the transcript matrix the color coded clustered were referred to as RNA clusters, while when the PCs were deduced from the lipid abundance intensities from the MSI matrix the cluster labeling was labeled as Lipid clusters. Lipid intensity clustering in

some naïve samples displays geometric patterns, which we attribute to incomplete spatial coverage during positive ionization mode acquisition. This reflects the inherent technical challenges of co-registering multiple ionization modes across multiple tissue sections, where complete spatial coverage in positive ionization mode was not achieved in some regions of these samples. The largest effect is seen in the naive sample, N4. In addition, our analytical method mitigates this by guiding the downstream analysis using RNA clusters with lipid intensities incorporated as metadata.

For sub-clustering only spots form three clusters (i.e., RNA_0/4/7) were subsetted. The sub-clustering was done using transcript data only and MSI data was retained as metadata. Single-cell RNA seq matrices were clustered using a reduced PC space and then clusters were checked for types by looking gene expression profiles by calculating DE per cluster identity by taking one versus all approach. Genes were then looked for canonical markers and cell types assigned.

The cell type information along with the single cell expression dataset and the Visium dataset was used as input for Cell2Location (v0.1) to model expression in single cells and then use that to deconvolute cell type abundances in space (Visium). The 5th quantile of the predicted abundances was used for all downstream analysis. Since the abundances were associated for each Vn this data is then easily integrated across the dataset. Correlations between lipid intensities and predicted cellular abundances were calculated using Pearson's correlation and the R function corr. Only statistically significant correlations were reported. Cell type abundances were compared between sub-clusters using a Kruskal-Wallis test with multiple comparisons to check for difference between groups. P-value of significance was set at 0.05.

## FACS sorted proteomics

Single-cell suspensions were stained in PBS with 0.05% BSA, using fluorescent antibodies against CD3, CD11b, F4/80, and CD172 (SIRPA). Macrophage populations (CD3–CD11b + F4/80 + ) were sorted on a Beckman Coulter MoFlo Astrios, separating cells based on CD172 expression into CD172hi and CD172lo subsets. Sorted cell populations were diluted 1:1 with aqueous 10% (v:v) sodium dodecyl sulfate, 100 mM triethylammonium bicarbonate (TEAB). Protein was reduced with 5.7 mM tris (2- carboxyethyl) phosphine and heated to 55°C for 15 min before alkylation with 22.7 mM methyl methanethiosulfonate at room temperature for 10 min. Protein was acidified with 6.5 µl of aqueous 27.5% (v:v) phosphoric acid then precipitated with dilution seven-fold into 100 mM TEAB90% (v:v) methanol. Precipitated protein was captured on S-trap (Profiti – C0-micro) and washed five times with 165 µl 100 mM TEAB 90% (v:v) methanol before digesting with the addition of 20 µl 0.1 µg/µl Promega Trypsin/Lys-C mix (V5071) in aqueous 50 mM TEAB and incubation at 47°C on hotplate for 2 h. Peptides were recovered from S-trap by spinning at 4000 g for 60 s. S-traps were washed with 40 µl aqueous 0.2% (v:v) formic acid and 40 µl 50% (v:v) acetonitrile:water and washes combined with the first peptide elution. Peptide solutions were dried in a vacuum concentrator then resuspended in 20 µl aqueous 0.1% (v:v) formic acid. Peptides were loaded onto EvoTip Pure tips for nanoUPLC using an EvoSep One system. A preset 30SPD gradient was used with a 15 cm EvoSep C18 Performance column (15 cm × 150 µm x 1.5 µm). The nanoUPLC system was interfaced to a timsTOF HT mass spectrometer (Bruker) with a CaptiveSpray ionisation source. Positive PASEF-DIA, nanoESI-MS and MS2 spectra were acquired using Compass HyStar software (version 6.2, Bruker). Instrument source settings were: capillary voltage, 1500 V; dry gas, 3 l/minute; dry temperature; 180 °C. Spectra were acquired between m/z 100-1,700. DIA windows were set to 25 Th width between m/z 400–1201 and a TIMS range of 1/K0 0.6–1.60 V.s/cm2. Collision energy was interpolated between 20 eV at 0.65 V.s/cm2 to 59 eV at 1.6 V.s/cm2.

LC-MS data, in Bruker .d format, was processed using DIA-NN (1.8.2.27) software and searched against and in-silico predicted spectral library, derived from the mouse subset of UniProt appended with common proteomic contaminants. Search criteria were set to maintain a false discovery rate (FDR) of 1%. High-precision quant-UMS[53] was used for

extraction of quantitative values within DIA-NN. Peptide-centric output in. tsv format, was pivoted to protein-centric summaries using KNIME 5.1.2 and data filtered to require protein q-values < 0.01 and a minimum of two peptides per accepted protein. Calculation of log2 fold difference and differential abundance testing was performed using limma via FragPipe-Analyst[54]. Sample minimum imputation was applied and the Hochberg and Benjamini approach was used for multiple test correction.

## Flow cytometry antibodies

Single-cell suspensions were incubated with fluorochrome-conjugated antibodies for 20 min at 4 °C in flow cytometry buffer (PBS with 2% FBS and 2 mM EDTA). The following antibodies were used: anti-mouse LPCAT2 polyclonal (ThermoFisher CL488-15082, Coralite488, 1:125), anti-mouse Ly6g (BioLegend 127616; clone 1A8, PerCP-Cy5.5, 1:100), anti-mouse F4/80 (BioLegend 123114; clone BM8, FITC, 1:100; Or PE-Cy7, 1:100), anti-mouse CD11b (ThermoFisher 17-0112-82; clone M1/70, APC, 1:200; Or BV421, 1:20), anti-mouse Ly6c (BioLegend 128032; clone HK1.4, BV421, 1:20), anti-mouse CD172a (ThermoFisher 12-1721-82; clone P84, PE, 1:100), anti-mouse CD3ε (BioLegend 100319; clone 145-2C11, PE-Cy7, 1:100), and anti-mouse NOS2 (ThermoFisher 61-5920-82; CXNFT, PE-eFluor 610, 1:300).

## Statistics and reproducibility

Data were analyzed with standard exploratory techniques as described under the sections *Single-cell RNA data pre-processing*, *Spatial Data Analysis*, *FACS Sorted Proteomics* and formal testing for statistical assumptions was not performed. Exact sample number definition is described under *Animals and Infection Model*.

## Reporting summary

Further information on research design is available in the Nature Portfolio Reporting Summary linked to this article.

## Data availability

Spatial and single-cell transcriptomic data available on gene expression omnibus with the accession codes GSE290324 and GSE290325 respectively. Proteomic mass spectrometry data sets and results files are referenced in ProteomeXchange (PXD058137) and available to download from MassIVE (MSV000096486) [https://doi.org/10.25345/C5ST7F84G]. Raw data for Fig. 6 (per granuloma cell detection) and Fig. 8 (FCS files, cell proportions and counts) are available at https://doi.org/10.6084/m9.figshare.30880784 and https://doi.org/10.6084/m9.figshare.30880754 respectively.

## Code availability

All related code and instructions are available on https://github.com/jipsi/spatial_lipid_gene with processed Rds files available on Zenodo[55].

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

## Acknowledgements

This work was funded by the York-Maastricht Partnership program and supported by a Wellcome Trust Investigator Award to PMK (WT224290). This research was part of the M4I research program and received financial support from the Dutch Province of Limburg under the LINK program. GM was supported by studentship from Department of Medical-Surgical and Transplant Pathophysiology and Department of Biomedical, Surgical and Dental Sciences, University of Milan. The authors would like to acknowledge the help of Dr Karen Hogg and Dr Karen Hodgkinson at the University of York's Bioscience Technology Facility I&C lab and that of Dr. Angelo Lopez and Dr. Chloë Baldreki within the University of York's Bioscience Technology Facility MAP-lab, for assistance with proteomic sample processing and LC-MS/MS data acquisition. The York Center of Excellence in Mass Spectrometry was created thanks to a major capital investment through Science City York, supported by Yorkshire Forward with funds from the Northern Way Initiative, and subsequent support from EPSRC (EP/K039660/1; EP/M028127/1). Figure 1a in this article has been created using Biorender.com.

## Author contributions

S.D., P.M.K., and R.H. conceived the study. S.D. and J.H.C. designed and performed the experiments. S.D. analysed the integrated data, created visualisations, and wrote the manuscript. B.B. and J.H.C. co-registered the images. S.D., H.A., and G.M. conducted the mouse experiments. S.D. and N.S.D. conducted the spatial transcriptomics. G.M. and S.D. conducted in vitro studies. S.J. and L.G. prepared libraries for sequencing from barcoded Visium spots. A.D. conducted proteomic data analysis. G.C., N.S.D., and P.O.T. provided insights and feedback. All authors edited the manuscript. P.M.K. and R.H. reviewed the manuscript and supervised the work.

## Competing interests

The authors declare no competing interests.
