## [Transparent Peer Review file · Communications Biology]

Ontogeny-independent expression of LPCAT2 in granuloma macrophages during experimental visceral leishmaniasis

Corresponding Author: Professor Paul Kaye

Version 0:

Reviewer comments:

Reviewer #1

(Remarks to the Author)

This manuscript presents a technically rigorous study integrating spatial transcriptomics, lipidomics, single-cell RNA-seq, and proteomics to characterize the immunometabolic landscape of hepatic granulomas during *Leishmania donovani* infection. The study is anchored on the novel finding of LPCAT2 expression in granuloma-associated macrophages and supports this conclusion with multilayered validation across modalities. This is a high-quality, comprehensive study with innovative methodology and biologically meaningful insights. Addressing the minor revisions listed below, and especially expanding the functional and translational discussion, would further elevate the manuscript.

Minor points for revision:

While the expression of LPCAT2 is convincingly demonstrated, the study does not address its functional role *in vivo*. Incorporating a discussion on potential LPCAT2 inhibition strategies or conditional knockout models (e.g., macrophage-specific deletion) would add valuable perspective and underscore the therapeutic relevance of this pathway. The discussion could benefit from a more critical appraisal of LPCAT2 translational potential, including possible risks, tissue specificity, and challenges in selective targeting.

The manuscript focuses on LPCAT2, but other enzymes involved in phospholipid remodeling (e.g., LPCAT1, LPCAT3) are only superficially mentioned. A detailed comparison of their expression patterns and potential roles in granulomatous inflammation would help contextualize the specificity and relevance of LPCAT2.

The authors report a weak correlation between parasite abundance and LPCAT2 expression. Were any differences observed in the lipid profiles (e.g., LPC/PC ratios or arachidonoylated species) between granulomas with high vs. low parasite loads? Also, it remains unclear whether LPCAT2 expression impacts inflammation resolution or granuloma involution during infection. A brief discussion on how these immunometabolic changes might influence disease outcome would be informative.

Could LPCAT2 expression be driven by systemic cytokines rather than local microenvironmental cues? Did the authors measure circulating cytokines or perform broader immunophenotyping to explore this? And, given that both Kupffer cells and MoMacs express LPCAT2, could the authors elaborate on other extrinsic factors that could drive this activation signature? This could help understand the stimulus-response specificity of LPCAT2 expression.

Were similar LPCAT2 expression patterns observed in other tissues (e.g., spleen)? Could this represent a generalizable mechanism across granulomatous diseases?

The figure legend for Figure 6 contains mislabeled panels (A–D), and panel 6C is not cited in the main text. Please revise the figure legend and ensure all subpanels are properly referenced and described.

Reviewer #2

(Remarks to the Author)

In their manuscript, Dey et al. employ an elegant multimodal approach including mass spectrometry-based lipid mapping,

spatial transcriptomics and conventional immunohistology to spatially analyze granuloma-associated in infected individuals compared to healthy individuals. By mapping the infection-associated changes lipid metabolism on the microenvironment during visceral Leishmaniasis, they identify Lysophosphatidylcholine acyltransferase LPCAT2 and their substrates to be important components of macrophage activation in *L. donovani* granulomas, both in Kupffer cells as well as in LysM+ Monocyte-derived Macrophages.

The metabolic, and in particular lipid, regulation of macrophages during infection is of high interest for understanding the deployment of efficient immune response. The manuscript presents an impressive characterization of the inflammatory microenvironment during *L. donovani* granuloma formation, employing state of the art spatial biology methods, and elegant analysis strategies. A functional validation of the importance of LPCAT2 for controlling the parasite, is largely missing. This would however greatly strengthen the manuscript, which, in its present form, rather represents a resource than a demonstration of the role of "LPCAT2 as a key player in the macrophage response during granulomatous inflammation", as the authors put it. Also, the manuscript contains some major mistakes in the figure preparation (e.g. missing panels), and suffers from a lack of clarity in the presentation of the findings, which we have discussed below.

Major comments:

1. The spatial biology investigation of LPCAT2 is admirable, however, the presented data do mainly provide correlative evidence of LPCAT2 expression with inflammatory activation. How does a manipulation of LPCAT2, its substrate, or any related pathway investigated in the manuscript, impact on the inflammatory activation of macrophages during *L. donovani* infection? Or is the differentially regulated lipid metabolism a result of the inflammatory activation?
2. In figure 1, the authors present a parallel analysis of lipidomic and transcriptomic spatial distribution. However, the claimed overlay of Lipid 4 cluster with RNA4 does not become evident, and seems to match as often as it deviates. The data presentation should be complemented with a more quantitative representation and statistical evaluation in order to back up the spatial organization.
3. In Figure 7, the authors show a comparison of cells sorted according to high versus low expression of SIRPα, which they use as a proxy for LPCAT2 which is intracellular and therefore not suitable for sorting of live cells. However the correlation of the markers is based solely on their transcriptomic data (Figure 4G), which does not necessarily overlap. Are LPCAT2 and SIRPα also correlated on the protein level? And is SIRPα also detectable in a similar distribution as LPCAT2 in the granuloma?
4. Figure 6 requires a major revision as the panel callouts in the main text, the figure description and the panels themselves do not match. To us it looked like the panel of the naïve mouse has been omitted and the images shown in A and B represents the same imaging side of a most likely infected animal. Also, the main message of panel C (Distribution of cell populations/granuloma, labeled "D" in the legend) remains hard to understand, how does it show that "50% of Kupffer cells expressed LPCAT2", as the authors state?

Minor comments

Figure 2 shows single cell RNAseq with the main results of changes in T/B cell proportion and increased MoMac as compared to Kupffer cells as a result of infection, which is not very surprising. The figure seems to be spread out a lot as compared to the limited new insights that it provides, and should be moved at least partially to the supplement.

Figure 5 displays the link between gene expression, cell abundance and lipids. However, it is hard to understand by non-spatial biology experts. Especially panel C in which we assume the spatial transcriptomic/lipidomic tiles are shown as individual dots. This is not explained in the figure legend.

In figure 6 B (labeled "C" in the legend), a segmentation is shown, however the segmentation is not covering the full image but only sub-areas. It did not become clear why the authors chose this partial segmentation.

In figure 7 A, the authors show an extended representation of their used gating strategy, which we recommend to show in detail in the supplements rather than in the main report. Also the underlying statistics of figure 7 C is not fully clear. Is this a paired analysis (which it should be, and which would probably make the differences in the lower graph significant)?

In figure S2, some of the distributions of the lipid spots look like they are very much related to sample preparation, as they follow geometric forms such as straight lines which might be due to cutting/preparation artifacts. Can the authors comment these "strange" distributions of lipid spots to non-experts in spatial lipidomics?

There is a typo in "elimination" in line 63.

There is a missing sentence separation in line 296.

Some abbreviations (such as AA for arachidonic acid) are not introduced at first occurrence, but later in the text.

(Remarks to the Author)

I co-reviewed this manuscript with one of the reviewers who provided the listed reports. This is part of the Communications Biology initiative to facilitate training in peer review and to provide appropriate recognition for Early Career Researchers who co-review manuscripts.

Version 1:

Reviewer comments:

Reviewer #1

(Remarks to the Author)

The authors have adequately addressed all of my comments and questions.

Reviewer #2

(Remarks to the Author)

The authors have performed extensive experimentation and have thoroughly revised the manuscript, including a new main figure regarding a possible functional relationship between LPCAT2 and iNOS expression in macrophages. A number of new figure panels and supplementary figures which make the manuscript better understandable and more convincing are also provided. This fully addresses the initial comments and we would like to congratulate the authors to their impressive work. I would like to recommend the publication of the manuscript without any reservation.

Reviewer #3

(Remarks to the Author)

I co-reviewed this manuscript with one of the reviewers who provided the listed reports. This is part of the Communications Biology initiative to facilitate training in peer review and to provide appropriate recognition for Early Career Researchers who co-review manuscripts.

Response to reviewers' comments (all changes in the manuscript are highlighted in yellow).

Reviewer #1

Remarks to the Author:

*This manuscript presents a technically rigorous study integrating spatial transcriptomics, lipidomics, single-cell RNA-seq, and proteomics to characterize the immunometabolic landscape of hepatic granulomas during *Leishmania donovani* infection. The study is anchored on the novel finding of LPCAT2 expression in granuloma-associated macrophages and supports this conclusion with multilayered validation across modalities. This is a high-quality, comprehensive study with innovative methodology and biologically meaningful insights. Addressing the minor revisions listed below, and especially expanding the functional and translational discussion, would further elevate the manuscript.*

We thank the reviewer for their appreciation of the technical robustness and novelty of our work.

Major points

The reviewer had no major points to address.

Minor points:

R1a: While the expression of LPCAT2 is convincingly demonstrated, the study does not address its functional role in vivo. Incorporating a discussion on potential LPCAT2 inhibition strategies or conditional knockout models (e.g., macrophage-specific deletion) would add valuable perspective and underscore the therapeutic relevance of this pathway. The discussion could benefit from a more critical appraisal of LPCAT2 translational potential, including possible risks, tissue specificity, and challenges in selective targeting.

We agree that we have not formally demonstrated a functional role for LPCAT2 in vivo, but have now included additional in vitro data that addresses the functional role of LPCAT2 (**new Fig. 8 and lines 263-287**). Regarding in vivo studies, while LPCAT2 knockout mouse models exist, they do not show obvious phenotypes. Conditional (floxed) LPCAT2 mice that would allow targeted deletion in myeloid cells are not currently available and hence are beyond the scope and timeline of the current manuscript. We have, as suggested, discussed their potential value in the Discussion (**new text, Lines 358-366**).

R1b: The manuscript focuses on LPCAT2, but other enzymes involved in phospholipid remodeling (e.g., LPCAT1, LPCAT3) are only superficially mentioned. A detailed comparison of their expression patterns and potential roles in granulomatous inflammation would help contextualize the specificity and relevance of LPCAT2.

The reviewer raises an important point. LPCAT2 is specific to RNA subcluster Sub0. We do see some LPCAT1 in Sub0 (Figure 5A) but is likely originating from B cells and APCs (Figure 3d). As suggested, we also now include data to show LPCAT1-4 are expressed in our ex vivo proteomic dataset, but only LPCAT2 is significantly differentially expressed in CD172^{high} vs. CD172^{low} cells (**new Fig. 7b, Lines 238-243**).

R1c: The authors report a weak correlation between parasite abundance and LPCAT2 expression. Were any differences observed in the lipid profiles (e.g., LPC/PC ratios or arachidonoylated species) between granulomas with high vs. low parasite loads? Also, it remains unclear whether LPCAT2 expression impacts inflammation resolution or granuloma involution during infection. A brief discussion on how these immunometabolic changes might influence disease outcome would be informative.

We acknowledge the value of lipid profiling granulomas with varying parasite loads, but we have not performed granuloma-level correlation analysis between specific lipid species (LPC/PC ratios or arachidonoylated species) and parasite abundance. This is due to the technical challenge of reliably identifying parasite transcripts in our spatial transcriptomic data. We have mentioned this limitation in the text (**new text, Lines 387-388**).

The weak correlation we observed between overall parasite abundance and LPCAT2 expression suggests that LPCAT2 upregulation may be driven more by the inflammatory microenvironment than by direct parasite burden. The reviewer raises an interesting point about the functional consequences of LPCAT2-mediated phospholipid remodeling on granuloma dynamics, which could not be formally answered without extensive longitudinal profiling spanning the entire course of infection. Nevertheless, our data showing co-expression of LPCAT2 with NOS2 and pro-inflammatory signatures and that LPCAT2 inhibition reduces nitric oxide production (**new Fig. 8**) suggests that LPCAT2 may help maintain the inflammatory state necessary for parasite containment (**new text, Lines 345-366**).

R1d: Could LPCAT2 expression be driven by systemic cytokines rather than local microenvironmental cues? Did the authors measure circulating cytokines or perform broader immunophenotyping to explore this? And, given that both Kupffer cells and MoMacs express LPCAT2, could the authors elaborate on other extrinsic factors that could drive this activation signature? This could help understand the stimulus-response specificity of LPCAT2 expression.

The reviewer raises an excellent question. While we did not measure circulating cytokines in this study, our spatial analysis provides several lines of evidence suggesting that LPCAT2 expression is primarily driven by local microenvironmental cues rather than systemic factors: i) LPCAT2 expression is highly localized to macrophages in granulomatous areas rather than all liver macrophages (Fig. 6), and ii) LPCAT2 expression correlates strongly with NOS2 expression (**new Fig. 8**) and is within granulomas, consistent with local activation (Figure 6A). Based on our data and literature, several local factors likely contribute to LPCAT2 upregulation in both Kupffer cells and monocyte-derived macrophages and these are now briefly discussed (**new text, lines 345-356**). For example, IFN γ and TNF within granulomas (evidenced by our transcriptomic data) likely contribute to increased LPCAT2 expression (**new Fig. 8**). In addition, *Leishmania* infection of macrophages may increase arachidonic acid metabolism (<https://doi.org/10.4049/jimmunol.134.1.556>), potentially creating a positive feedback loop for LPCAT2 expression.

R1e: Were similar LPCAT2 expression patterns observed in other tissues (e.g., spleen)? Could this represent a generalizable mechanism across granulomatous diseases?

We thank the reviewer for this suggestion. To check this, we used a publicly available dataset of single-cell RNA sequencing of spleen and bone marrow from *L. donovani* infected BALB/c mice (<https://doi.org/10.1016/j.celrep.2023.113097>). We visualised these two tissues in reduced UMAP dimensions and generated feature plots for *Lpcat2* and the macrophage marker *Csf1r*. We found a small population of *Csf1r*⁺ *Lpcat2*⁺ cells in both spleen and bone marrow (**new Supplementary Fig. S5b; new text, lines 166-168**). However, it should be noted that neither spleen nor bone marrow develop granulomatous inflammation in this infection model, limiting generalisability.

R1f: The figure legend for Figure 6 contains mislabeled panels (A–D), and panel 6C is not cited in the main text. Please revise the figure legend and ensure all subpanels are properly referenced and described.

We acknowledge this oversight and have now updated the legends. Figure 6 has been extensively revised to include new F4/80, LPCAT2, SIRP α and NOS2 staining (**new Fig. 6a**). Additional figure panel (**new Fig. 6e**) has been added to clarify granuloma composition with respect to ApoE and LPCAT2 expression.

Reviewer #2

Remarks to the Author

*In their manuscript, Dey et al. employ an elegant multimodal approach including mass spectrometry-based lipid mapping, spatial transcriptomics and conventional immunohistology to spatially analyze granuloma-associated in infected individuals compared to healthy individuals. By mapping the infection-associated changes lipid metabolism on the microenvironment during visceral Leishmaniasis, they identify Lysophosphatidylcholine acyltransferase LPCAT2 and their substrates to be important components of macrophage activation in *L. donovani* granulomas, both in Kupffer cells as well as in LysM⁺ Monocyte-derived Macrophages.*

*The metabolic, and in particular lipid, regulation of macrophages during infection is of high interest for understanding the deployment of efficient immune response. The manuscript presents an impressive characterization of the inflammatory microenvironment during *L. donovani* granuloma formation, employing state of the art spatial biology methods, and elegant analysis strategies. A functional validation of the importance of LPCAT2 for controlling the parasite, is largely missing. This would however greatly strengthen the manuscript, which, in its present form, rather represents a resource than a demonstration of the role of “LPCAT2 as a key player in the macrophage response during granulomatous inflammation”, as the authors put it. Also, the manuscript contains some major mistakes in the figure preparation (e.g. missing panels), and suffers from a lack of clarity in the presentation of the findings, which we have discussed below.*

We thank the reviewers for their kind comments and their appreciation of the depth and quality of our study.

Major comments:

*R2a: The spatial biology investigation of LPCAT2 is admirable, however, the presented data are do mainly provide correlative evidence of LPCAT2 expression with inflammatory activation. How does a manipulation of LPCAT2, its substrate, or any related pathway investigated in the manuscript, impact on the inflammatory activation of macrophages during *L. donovani* infection? Or is the differentially regulated lipid metabolism a result of the inflammatory activation?*

We appreciate this important question about functional validation. The lack of genetic tools or drugs for in vivo inhibition of LPCAT2 precludes addressing this question directly (see also comments to R1). To address this issue therefore, we conducted *in vitro* functional studies using bone marrow-derived macrophages and the selective LPCAT2 inhibitor TSI-01. Our results demonstrate that LPCAT2 inhibition significantly reduces NOS2 expression and nitrite production in LPS/IFN- γ stimulated macrophages, providing direct evidence of a role for LPCAT2 in inflammatory activation and in the induction of a key anti-leishmanial effector mechanism (**new Fig. 8, Lines 263-287**). Since TSI-01 acts through competitive inhibition of arachidonoyl-CoA incorporation, this supports the hypothesis that arachidonic acid metabolism via LPCAT2 is functionally important for mounting effective inflammatory responses during granulomatous inflammation rather than simply being a downstream consequence of inflammatory activation.

R2b: In figure 1, the authors present a parallel analysis of lipidomic and transcriptomic spatial distribution. However, the claimed overlay of Lipid 4 cluster with RNA4 does not become evident and seems to match as often as it deviates. The data presentation should be complemented with a more quantitative representation and statistical evaluation to back up the spatial organization.

We thank the reviewer for this comment and have now added a figure that depicts the percentage overlap between RNA and Lipid clusters (**new Supplementary Fig. S1**). We focussed on Lipid_4 and RNA_4 for the following reasons: i) Both RNA_4 and Lipid_4 spots are increased upon infection (Fig. 1e); ii) These clusters overlap the most (**new Supplementary Figure S1, Lines 105-106**) and overlap with granulomas (Fig. 1f); and iii) RNA_4 clusters show a distinct immune activation signature (Supplementary Fig. S3a). While the lipid clusters show that lipids are spatially organised, our analysis from Figure 4 onwards uses only RNA clusters (RNA_0/4/7) that are predicted to contain immune cells (Fig. 4a) for further characterisation (Fig. 4b and Fig. 5).

R2c: In Figure 7, the authors show a comparison of cells sorted according to high versus low expression of SIRPa. which they use as a proxy for LPCAT2 which is intracellular and therefore not suitable for sorting of live cells. However the correlation of the markers is based solely on their transcriptomic data (Figure 4G), which does not necessarily overlap. Are LPCAT2 and SIRPa also correlated on the protein level? And is SIRPa also detectable in a similar distribution as LPCAT2 in the granulome?

This is an important point raised by the reviewer. In Fig. 7b, we show that SIRP α high sorted cells have higher LPCAT2 based on proteomics. However, we have now included additional protein expression data to show colocalization of SIRP α , LPCAT2 and NOS2 in granulomas using immunohistochemistry (**new Fig. 6a and Supplementary Fig. S7b; Lines 218-222; 555-560**).

R2d: Figure 6 requires a major revision as the panel callouts in the main text, the figure description and the panels themselves do not match. To us it looked like the panel of the naïve mouse has been omitted and the images shown in A and B represents the same imaging side of a most likely infected animal. Also, the main message of panel C (Distribution of cell populations/granuloma, labeled “D” in the legend) remains hard to understand, how does it show that “50% of Kupffer cells expressed LPCAT2”, as the authors state?

We thank the reviewers for raising these points. The naïve mouse data panel was part of an earlier iteration of the figure that was later removed as naïve tissue did not show any LPCAT2 expression. We have removed this text from the figure legend. We also agree that the original panel C/D “Distribution of cell populations / granuloma” did not directly show that 50% of Kupffer cells expressed LPCAT2 but rather shows a distribution of cells within a granuloma.

We have now fully revised Fig. 6 (**new Fig. 6**). Panel a and b show representative granuloma images from two immunostaining panels. Single channel images are provided in **new Supplementary Fig. S7**. Panel c now shows the analysis strategy to obtain masks and per-cell intensity information used to infer panels d-f. Panel d shows the percentage of Lpcat2+, ApoE+, Lpcat2+ ApoE+ and double negative cells in each granuloma split by mouse (n=234 granulomas). Panel e shows the breakdown of all myeloid cells (based on ApoE/LPCAT2 staining) across all granulomas (n=15,701 cells). Panel f shows per granuloma occurrence of ApoE+ cells that are also LPCAT2+ (n=234 granulomas). Panel g shows correlation between LPCAT2+ cells and parasite presence (OpB+) per granuloma (n=234 granulomas). Corresponding text changes have been made to the Results describing Figure 6 (**Lines 217-228**)

Minor comments

R2e: Figure 2 shows single cell RNAseq with the main results of changes in T/B cell proportion and increased MoMac as compared to Kupffer cells because of infection, which is not very surprising. The figure seems to be spread out a lot as compared to the limited new insights that it provides and should be moved at least partially to the supplement.

Whilst we agree in part with the reviewer’s comment, we wish to maintain the content of Fig. 2. In addition to being one of the first single-cell RNAseq datasets available for *Leishmania* infected liver, the data provides novel information on T cell sub-clusters in infected mice, notably with regard to *Ifng*. This directly relates to issues discussed later in the manuscript (Supplementary Fig. S4a-d), and in conjunction with the deposited scRNA-seq data set provides a valuable resource for the research community.

R2f: Figure 5 displays the link between gene expression, cell abundance and lipids. However, it is hard to understand by non-spatial biology experts. Especially panel C in which we assume the spatial transcriptomic/lipidomic tiles are shown as individual dots. This is not explained in the figure legend.

We have now added text in Fig. 5 legend describing that each dot in the violin plots represents a spatial Visium spot (**new Figure 5 legend**).

R2g: In figure 6 B (labeled “C” in the legend), a segmentation is shown, however the

segmentation is not covering the full image but only sub-areas. It did not become clear why the authors chose this partial segmentation.

Thank you for pointing this out. We have added this information in the methods under immunohistochemistry (**Lines 562-565**). Briefly, we selected granulomas manually to ensure segmentation is carried out only within granulomas (234 granulomas across 4 infected mice) to obtain per granuloma cell distribution of ApoE⁺ and/or LPCAT2⁺ cells.

R2h: In figure 7 A, the authors show an extended representation of their used gating strategy, which we recommend to show in detail in the supplements rather than in the main report. Also the underlying statistics of figure 7 C is not fully clear. Is this a paired analysis (which it should be, and which would probably make the differences in the lower graph significant)?

As requested, we have now moved the old Fig. 7A to supplementary (**new Supplementary Fig. S8**). We have added text (Lines 1056-1057) describing the statistics used for proteomics in the figure legend and cross-referenced to Methods, where this is explained in more detail. We have unequal sample sizes in this analysis due to sample loss post-sorting (CD172_{hi}; n=8 and CD172_{low}; n=5, as described in the legend) which does not allow for paired analysis (**Lines 427-430**).

R2i: In figure S2, some of the distributions of the lipid spots look like they are very much related to sample preparation, as they follow geometric forms such as straight lines which might be due to cutting/preparation artifacts. Can the authors comment these “strange” distributions of lipid spots to non-experts in spatial lipidomics?

We thank the reviewer for this observation. The geometric patterns in certain samples (especially naive N4) result from incomplete spatial coverage in positive ionization mode acquisition. Importantly, our analytical approach addresses this by using RNA-based clustering to guide downstream analysis, ensuring these artifacts do not impact our primary findings. We now explicitly mention this in the revised Results and Methods sections (**Lines 112-116; 605-611**).

R2j: There is a typo in “elimination” in line 63.

This is now corrected, thank you.

R2k: There is a missing sentence separation in line 296.

Thank you. Sentence separation now added.

R2l: Some abbreviations (such as AA for arachidonic acid) are not introduced at first occurrence, but later in the text.

These are now introduced appropriately, thank you.